# Development and initial testing of a brief, generic self-reported disability questionnaire: The Universal Disability Index

David William Evans[1,2]*

1 School of Sport, Exercise and Rehabilitation Sciences, University of Birmingham, Birmingham, United Kingdom, 2 University College of Osteopathy, London, United Kingdom

* dwe@backpainclinic.co.uk

## Abstract

**Data Availability Statement:** Anonymised data are available from: https://figshare.com/projects/Development_of_Universal_Disability_Index/195955.

### Background

Disability is an important multifaceted construct. A brief, generic self-reported disability questionnaire that promises a broader and more comparable measure of disability than disease-specific instruments does not currently exist. The aim of this study was to develop and evaluate such a questionnaire: the Universal Disability Index (UDI).

### Methods

An online survey was used to collect general population data. Data were randomly divided into training and validation subsets. The dimensionality and structure of eight UDI questionnaire items were evaluated using exploratory factor analysis (EFA, training subset) followed by confirmatory factor analysis (CFA, validation subset). To assess concurrent validity, the UDI summed score from the full dataset was compared to the Groningen Activity Restriction Scale (GARS) and the Graded Chronic Pain Scale (GCPS) disability scores. Internal consistency and discriminant validity were also assessed. Bootstrapping was used to evaluate model stability and generalisability.

### Results

403 participants enrolled; 364 completed at least one UDI item. Three single-factor versions of the UDI were assessed (8-item, 7-item, and 6-item). All versions performed well during EFA and CFA (182 cases assigned to each), but none met the RMSEA (Root Mean Square Error of Approximation) criterion ($\leq 0.08$). All versions of the UDI had high internal consistency (Cronbach's $\alpha > 0.90$), were strongly correlated (Pearson's $r > 0.7$) with both GARS and GCPS disability scores, indicating concurrent validity, and could accurately discriminate between upper and lower quartiles of these comparators. Confidence intervals of estimates were narrow, suggesting model stability and generalisability.

**Funding:** The author(s) received no specific funding for this work.

**Competing interests:** The authors have declared that no competing interests exist.

## Conclusions

A brief, generic self-reported disability questionnaire was found to be valid and to possess good psychometric properties. The UDI has a single factor structure and either a 6-item, 7-item or 8-item version can be used to measure disability. For brevity and parsimony, the 6-item UDI is recommended, but further testing of all versions is warranted.

## Introduction

Disability is a multifaceted construct with enormous implications for individual well-being, healthcare systems, and societal burden. In the United States alone, at least 1 in 4 adults are living with a disability [1] on a daily basis. A similar proportion is mirrored in other countries [2–4], highlighting the universal importance and relevance of disability.

Contemporary definitions and understanding of disability have evolved over time to transcend specific medical conditions, encompassing a broader context that integrates the interplay of biological, psychological, and socio-environmental factors. Accordingly, the World Health Organization's International Classification of Functioning, Disability and Health (ICF) [5] suggests a comprehensive and globally accepted biopsychosocial framework for understanding and assessing disability with parity across all health conditions, physical and psychological. According to the ICF, disability manifests as activity limitations and participation restrictions, thereby placing activities of daily living (ADLs) [6] at the centre of disability assessment.

Accurate and comprehensive measurement of disability is paramount for clinical diagnosis, planning patient management, and public health interventions. Disability assessments, whether based on in-person assessments such as standardised observations [7–16] and interviews [17, 18], or self-reported questionnaires [11, 19–21], uniformly centre around ADLs. However, in-person assessments are time-consuming (typically up to 60 minutes), expensive, and can even be distressing for the person being assessed [22]. If the assessed has very severe levels of disability, in-person assessment may be the only option. However, people with all but the most severe levels of disability are often able to self-report their capabilities and restrictions through questionnaires. Given the broad range of communication technologies now available, self-reported questionnaires offer the benefit of remote, asynchronous disability assessment, which is attractive for (1) higher frequency testing, (2) limiting the spread of communicable disease, and (3) reducing unnecessary travel burden on individuals with mobility problems. Despite this, most generic measures of disability are not self-reported and instead rely on in-person observation.

Unlike the WHO's broad view of disability, encapsulated within the ICF [5], self-reported disability questionnaires are typically specific to a given disease or condition. While disease-specific assessments (e.g., [23–28]) undoubtedly offer in-depth insights into nuances of particular conditions, they inherently limit the breadth of information captured about the individual's level of functioning, presumably under the assumption that some of this information is irrelevant. The use of multiple disease-specific disability questionnaires also hinder comparisons between different conditions, in the same way that generic quality of life measures currently allow [29]. These inbuilt limitations of disease-specific measures underscore the potential advantages of employing a generic, non-attributed measure of disability, which potentially offers a broader view of a person's capabilities and restrictions. Indeed, disability measures are central to evaluating the therapeutic effectiveness of many interventions within

clinical trials [30–32]. But in complex interventions [33], the source and nature of therapeutic effect is not always clear or might be incorrectly attributed. Hence, in such interventions, a generic measure of disability is more likely to capture a therapeutic effect if one exists. Such an instrument would also be valuable if able to distinguish between individuals who are truly disabled and those who are not, irrespective of whether they have received a medical diagnosis and therefore qualify for assessment via a disease-specific measure. Surprisingly, relatively few generic self-reported disability questionnaires are currently available. Of those that are, there is either a focus on a specific aspect of disability, such as task independence [19], they incorporate complex 'instrumental' activities that may not be universally relevant or might be outdated [11, 19, 34], or they are very lengthy [11].

With these considerations in mind, the primary aim of this study was to develop and evaluate the performance of a brief, generic self-reported disability questionnaire. This instrument seeks to address an existing gap in the literature by providing a versatile, yet rigorous, tool for assessing ADL-related disability across various demographic settings and health conditions.

## Methods

### Data collection

Prior to recruitment and data collection, ethical approval was gained from the Research Ethics Committee of the School of Sport, Exercise and Rehabilitation Sciences, University of Birmingham (ref: MCR2223_08). All data were collected within an open online (web browser) survey between 22nd November 2022 and 28th August 2023, created using REDCap data collection software [35] and hosted on protected university servers.

Access to the online survey was gained via a public web link (URL). The URL directed interested individuals to an online participant information page containing further details of the study. Participation was voluntary with no incentives or remunerations offered or provided for taking part. Once respondents confirmed that they had read and understood the study information, they could proceed to complete an online self-declared eligibility form.

### Participants

Convenience sampling (e.g., email, word of mouth, social media posts) was used to recruit members of the general public. To encourage a diverse sample of respondents, invitations were posted on social media groups representing geographical regional networks and support groups for conditions commonly associated with high levels of disability (e.g. back pain, sciatica, chronic pain, fibromyalgia, chronic fatigue syndrome, depression and anxiety, etc.).

Eligibility criteria were designed to encourage broad participation, including healthy participants as well as those with disabling conditions. Hence, inclusion criteria only required participants to be aged 18 years or more and able to understand English to follow instructions and understand questions. No other exclusion criteria were used. Self-declared eligibility permitted advancement to the online consent form. Once signed and timestamped consent was gained, the participant was presented with the first page of the online survey.

### Survey contents

The online survey was conducted according to best practice standards, according to the Checklist for Reporting Results of Internet e-Surveys (CHERRIES) framework [36]. This included piloting and field testing the survey to optimise the data collection process while minimising burden on respondents. The online survey consisted of a series of questionnaires, each of which was displayed in a separate page within participant's web browsers:

## Demographics

Demographic details were collected to help describe participants and check that the sample was sufficiently diverse. These details included current age, sex-at-birth, country of residence, ethnicity, educational attainment, current employment status, and smoking status.

## Groningen Activity Restriction Scale

The Groningen Activity Restriction Scale (GARS) [19, 37, 38] is a non-disease specific scale used to determine a respondent's current independence when attempting to perform ADLs. It consists of 18 items, each of which represents a different ADL and is scored on a 5-point ordinal scale from 1 to 5, where higher scores indicate greater self-reported disability. A total summed score can therefore be derived with minimum and maximum scores of 18 and 90 respectively. The GARS has been shown to possess good psychometric properties [38].

## Graded Chronic Pain Scale (4-week recall)

Given that pain is highly prevalent in the general population [39, 40], it was deemed essential that a measure of pain-related disability was included. The Graded Chronic Pain Scale (GCPS) [41] is a widely used, brief measure of pain severity that can be used for pain at any anatomical location. The 4-week recall version of the GCPS [42] was used in this study. Three items measure different variants of pain intensity (*current*, *average*, and *worst* pain) on 11-point (0–10) numerical rating scales in which a higher score represents greater pain intensity. The 'characteristic' pain intensity score (range: 0–100) is obtained by calculating the mean of these three pain intensity scales and multiplying this by 10. The GCPS is also comprised of three further 11-point numerical rating scales measuring disability in terms of pain interference (*daily activities*, *social activities*, and *work activities*) with higher scores representing greater pain interference. The disability score (range: 0–100) is obtained through calculating the mean of these pain interference scales and multiplying this by 10.

## Universal Disability Index

The newly developed scale, the Universal Disability Index (UDI), was designed to measure self-reported restriction of the respondent's ability to perform eight important ADLs: *walking, standing, sitting, lifting and carrying, work and daily routine, washing and dressing, sleeping,* and *social and recreational activities*.

   The UDI was informed by and partially derived from existing, highly regarded disability questionnaires, primarily the Oswestry Disability Index, version 2.1a (ODI) [43, 44] and the Neck Disability Index (NDI) [45], which share a similar format (the latter was based on the former). As such, the ADLs covered by the UDI encompassed seven that are assessed within the ODI (*personal care*, *lifting*, *walking*, *sitting*, *standing*, *sleeping*, and *social life*) and one assessed within the NDI (*work*). Similar to the ODI and NDI, the six ordinal response options accompanying each UDI8 item present concrete examples of increasing restriction of a single ADL that are scored from 0 to 5 respectively (see supporting information for full wording). Concrete examples of ADL ability were chosen to ensure consistency between levels of reported (dis)ability, rather than asking respondents to gauge their recent ability compared to their historical levels, which would inevitably vary between individuals. The total UDI score was calculated by summing the scores of all completed items and multiplying by a factor of two. For the eight UDI items, a minimum score of 0 and maximum score of 80 was therefore achievable.

Although *pain* is an item in both the ODI and NDI, a *pain* item was not incorporated into the UDI because of the desire to create a condition-generic questionnaire based on ADLs. Instead, items related to everyday activities (e.g., walking, standing, lifting) or life situations (e.g., social, recreational, or work activities) were chosen so that the UDI aligned with the WHO's ICF framework [5]. Similarly, all eight UDI items were deliberately worded to be devoid of any attribution to a specific source of disability (e.g., pain, fatigue, visual impairment, etc.). A recall period of two weeks was implemented to provide respondents with a well-defined timeframe that would maximise the likelihood of accurate recollection but also capture a period of time that would allow for symptom fluctuations. As such, each question of the UDI was worded in the form: "*How much have you been able to [perform the ADL] during the past 2 weeks*?" This wording was chosen since asking participants whether they have "been able to" perform the ADL was considered to represent *ability*. By comparison, asking a person how much they have actually performed an ADL over a given period was considered to also incorporate opportunity. For instance, during recent COVID-19 lockdowns, an individual might have possessed the physical ability to walk long distances yet might answer negatively when asked the question "*How much have you walked during the past 2 weeks*?", because imposed legal restrictions will have removed their opportunity to exercise their ability to walk more than a short distance.

## Statistical analysis

R statistical software, version 4.3.1 [46] was used for all data processing and analyses. Demographics and clinical features (GARS total score, GCPS disability score, GCPS pain intensity score, etc.) were summarised descriptively for all participants.

## Missing data and response distributions

UDI responses were evaluated for missing values. Any cases in which all UDI values were missing were removed from all subsequent analyses beyond describing participant characteristics. Of the retained cases, missing UDI data were tested using Little's missing completely at random (MCAR) test [47], as implemented within the 'naniar' package for R. Since the distribution of UDI data was not known, missing UDI values were then imputed using the K-Nearest Neighbour algorithm [48] (with $k = 5$), via the 'VIM' R package, which does not assume a particular distribution. Floor and ceiling effects (defined as $\geq$50% of respondents selecting the lowest or highest option respectively) were assessed before and after imputation.

## Internal consistency

Following imputation, the internal consistency of the UDI and all other instruments was estimated on the full sample through calculation of Cronbach's $\alpha$. Ninety-five percent confidence intervals (95% CIs) for Cronbach's alpha were estimated using Duhachek's method [49], which does not assume item intercorrelations follow a normal distribution.

## Data suitability for factor analysis

The dimensionality of the data and underlying structure of the UDI was investigated using factor analysis. To ensure the data were suitable, Bartlett's Test of Sphericity and the Kaiser-Meyer-Olkin (KMO) Measure of Sampling Adequacy were used to assess the suitability of the data for factor analysis. Bartlett's Test is used to test the null hypothesis that the correlation matrix is an identity matrix, which would indicate that the variables are unrelated and therefore unsuitable for structure detection through factor analysis. A significant *p*-value ($< 0.05$)

for Bartlett's Test indicates that there are some relationships between variables, and thus the data is likely suitable for factor analysis. The KMO measure is an index, ranging from 0 to 1, used to examine the proportion of variance among the variables that might be common variance. A KMO value close to 1 indicates that the patterns of correlations are relatively compact, and factor analysis should yield distinct and reliable factors. Generally, a KMO value above 0.6 is considered adequate for factor analysis.

## Sub-setting data

Depending on the size of the available sample, at least 20 cases per UDI item [50] (i.e., at least 160 cases) were to be selected for a training dataset for exploratory factor analysis (EFA). This number was also anticipated to be necessary for a validation dataset to perform confirmatory factor analysis (CFA) [51]. Hence, if the study sample was sufficiently large (i.e., 320 or more useable responses), the sample would be randomly divided into training and validation subsets; otherwise, the full dataset would be used only for EFA, and CFA would not be performed. In the event of data sub-setting, the distribution of participant characteristics between the two subsets would be compared.

## Number of factors

The determination of the number of factors to extract was guided by four different statistical methods, each independently applied to the training data set with a bootstrap sampling technique that incorporated 5,000 iterations. For each method, the distribution of retained factor numbers was plotted in a histogram to aid interpretability. Firstly, a bootstrapped Parallel Analysis (using the 'fa.parallel' function from the 'psych' R package) was conducted on the training dataset, generating eigenvalues from the correlation matrix and averaging these across the bootstrap iterations. In each iteration, the calculated mean was contrasted with the mean derived from uncorrelated random data. Factors were retained within each iteration if their averaged eigenvalues exceeded those of the random data. Secondly, the Comparative Data approach of Ruscio and Roche [52] was used (via the 'CD' function implemented in the 'EFA-tools' R package). For each iteration, this method used resampled subsets of the training dataset to generate a range of possible factor solutions. Eigenvalues derived from the actual data were compared against those from the simulated data. Thirdly, the Minimum Average Partial (MAP) method [53] was used (using the 'VSS' function from the 'psych' R package). This method systematically reduces shared variance among components until only unique variance remains. Finally, mean eigenvalues with corresponding 95% CIs were calculated, the number with a value above 1.0 were counted, and a scree plot was created. In the event that the number of factors to be extracted was not clearly determined by these four bootstrapped methods, multiple factor analyses would be performed and compared.

## Exploratory factor analysis

Once a decision had been made on the number of factors to be extracted, bootstrapped exploratory factor analysis (EFA) was performed on the training dataset. Bootstrapping the EFA provides more robust estimates of factor loadings and error variances, along with 95% CIs, enhancing the stability and reliability of the factor solution, especially in the presence of a small sample size or non-normal data [54]. To implement this, an initial EFA was conducted on the training dataset, utilising the chosen number of factors (as detailed above) to construct a 'target' matrix (i.e., factor structure). Since a normal distribution of responses across each variable was not assumed, and factors were likely to be correlated, this initial EFA was

performed using principal axis factor extraction with direct oblimin rotation (via the 'fa' function of the 'psych' R package).

Once an initial factor structure had been created, a bootstrapped EFA with 5,000 iterations (again using principal axis factor extraction with direct oblimin rotation on each iteration) was then used to obtain mean factor loadings, communalities, eigenvalues, and standardised residuals, accompanied by their respective 95% CIs. Since factors may vary in their orders across EFA iterations, if a solution required more than one factor, Procrustes rotation (using the 'PROCRUSTES' function from the EFA.dimensions R package) would be utilised to ensure that all factor loadings were located in a common factor space (i.e., relative to the initial factor structure). This step would align all bootstrapped EFA iterations with the initial factor structure to make sure that 'Factor $n$' of iteration 1 would be the same as 'Factor $n$' in all other iterations [54].

The 95% CIs provided by bootstrapping the EFA models were utilised for judging their robustness. If a model was not deemed robust, items would be removed and the EFA performed again. A bootstrapped EFA model was considered robust if the 95% CIs of factor loadings did not include a value of 0.32 or below (i.e., loadings were not weak) [55]. Similarly, in a multi-factor model, an item would be considered robust if its 95% CIs for loadings did not include a value of 0.32 or above on more than one factor (i.e., low cross-loading). The intention behind these thresholds was to ensure each item primarily contributed to a single factor and that each factor had a significant relationship with its associated items [50, 55]. Communalities at extraction from the bootstrapped EFA results were computed separately for each bootstrap iteration as the sum of squared factor loadings across all factors for each item. Any item with a communality 95% CI that included 0.40 or less was considered for exclusion [56]. A model was deemed robust if a minimum of 50% of the total scale variance could be accounted for by the included items. Standardised 'non-redundant' residuals for each item were inspected to assess EFA model fit, since large residuals indicate a poor fit between observed and predicted correlations [57]. Specifically, if the upper bound of the 95% CIs of the standardised residuals exceeded a commonly used 0.5 threshold [57] in more than half of the UDI items, this would suggest potential issues with the factor structure or the need for further refinement of the model.

### Confirmatory factor analysis

Following the bootstrapped EFA, on the assumption of sufficient participant numbers, remaining cases were utilised as the validation dataset for Confirmatory Factor Analysis (CFA), during which the items and factor structure derived from the EFA was evaluated. Bootstrapping, again with 5,000 iterations, was utilised to estimate model stability, with the use of the 'bootstrapLavaan' function of the 'Lavaan' R package. Fit indices (and their accompanying 95% CIs) were used to evaluate the bootstrapped CFA against conventionally acceptable values [58, 59]: chi-square test $p > 0.05$, comparative fit index (CFI) $> 0.90$, Tucker-Lewis index (TLI) $> 0.90$, root mean square error of approximation (RMSEA) $< 0.08$, and standardised root mean square residual (SRMR) $< 0.05$. In addition, the mean and 95% CIs of Akaike Information Criterion (AIC) Bayesian Information Criterion goodness-of-fit statistics, for which lower values represent a better model fit, were calculated so that multiple CFA models could be compared.

### Concurrent validation of the UDI

Concurrent validity of the UDI was evaluated by inspection of Pearson's correlation coefficient $r$ with established comparator disability scales, the GARS and the disability score of the GCPS.

Again, bootstrapping was used to calculate point estimates (means) and 95% CIs for correlation coefficients. While the GARS (which measures independence in the performance of ADLs) and GCPS (which measures pain interference during ADLs) gauge disability in different ways to the UDI, it was hypothesised that correlations between the UDI and the comparator disability measures should *at minimum* be in the same direction (i.e., positive) and reach the conventional level of statistical significance (i.e., a *p*-value < 0.05, calculated using Fisher's Z transformation). Moreover, a strong correlation between the UDI and these comparator measures (i.e., point estimate of Pearson's *r* > 0.7) was expected. Inclusion of the UDI point estimate within the 95% CI of these comparator disability measures would further strengthen confidence in concurrent validity. The full dataset was used for these comparisons (i.e., all subsets combined). In addition, as a sensitivity analysis, correlations were also assessed in the subset of participants with a non-zero 'current' pain intensity score (one of the GCPS items). Prior to these comparisons, missing GARS and GCPS values were imputed using the K-Nearest Neighbour algorithm (with *k* = 5), via the 'VIM' R package. If any UDI items were removed during the prior factor analyses, all 'versions' of the UDI would be compared against the comparator instruments.

### Discriminant validity of the UDI

The ability of the UDI to discriminate between individuals with low and high levels of disability was also evaluated as a form of external validity. Firstly, the upper and lower quartiles of every disability scale were tested against each another using Mann Whitney U to ensure they were significantly different. This provided a form of internal validation to establish that each scale possessed sufficient variability in the current dataset to discriminate between different levels of the underlying constructs they are purported to measure. UDI total scores were then used to discriminate between upper and lower quartiles of comparator variables using ROC curves. This provided several measures of discriminatory ability for each comparison, such as area under the curve (AUC), sensitivity, and specificity, along with an optimal threshold UDI score.

## Results

### Participants

Five-hundred and twelve individuals responded to the survey invitation. Of these, 492 declared themselves eligible to participate, and 403 completed written consent to participate. One participant requested their data be completely withdrawn, without giving reason, but having not completed all instruments. Demographic details of the remaining 402 participants are displayed in Table 1. Briefly, the mean age of participants was 44.7 (SD: 17.4) years and the majority were female (280/402, 69.7%) and white (364/402, 90.5%). Most participants resided in the United Kingdom (294/402, 69.65%), although there was some geographical diversity achieved. The health-related data indicated that the sample appeared to be reasonably health diverse with more than a third either being a current smoker or having smoked previously, and 69.9% (281/402) experiencing some pain (above zero on a 0–10 scale) at the time of responding. Notably, the mean UDI total score was at the lower end of its range at 18.55 (SD: 18.41) from a possible 80 points.

### Missing data and response distributions

Of the 402 participants, 334 (83.1%) completed all eight UDI items, while 364 (90.5%) completed at least one. Thus, 38 cases in which all UDI data were missing were removed for all

**Table 1. Participant characteristics.**

| Characteristic | Value |
|---|---|
| Age, mean (SD) | 44.7 (17.4) |
| Sex assigned at birth, n (%) | Female, 280 (69.7) |
| | Male, 115 (28.6) |
| | Prefer not to say, 2 (0.5) |
| | Missing, 5 (1.2) |
| Country of residence, n (%) | United Kingdom, 294 (73.1) |
| | United States of America, 38 (9.5) |
| | Australia, 27 (6.7) |
| | Cyprus, 10 (2.5) |
| | Canada, 8 (2.0) |
| | Republic of Ireland, 5 (1.2) |
| | Greece, 5 (1.2) |
| | France, 2 (0.5) |
| | Italy, 2 (0.5) |
| | Switzerland, 2 (0.5) |
| | New Zealand, 1 (0.2) |
| | Oman, 1 (0.2) |
| | Portugal, 1 (0.2) |
| | United Arab Emirates, 1 (0.2) |
| | Missing, 5 (1.2) |
| Ethnicity, n (%) | White, 364 (90.5) |
| | Asian or Asian British, 6 (1.5) |
| | Black or Black British, 6 (1.5) |
| | Chinese or Chinese British, 4 (1.0) |
| | Another ethnic group, 14 (3.5) |
| | Prefer not to say, 2 (0.5) |
| | Missing, 5 (1.2) |
| Left full-time education, n (%) | Aged 16 or less, 43 (10.7) |
| | Aged 17–19, 66 (16.4) |
| | Aged 20 or over, 225 (56.0) |
| | Still in full-time education, 62 (15.4) |
| | Preferred not to say, 1 (0.2) |
| | Missing, 5 (1.2) |
| Work, n (%) | Currently working, 226 (56.2) |
| | Currently not working, 164 (40.8) |
| | Preferred not to say, 7 (1.7) |
| | Missing, 5 (1.2) |
| Smoker, n (%) | Current, 43 (10.7) |
| | Previous, 111 (27.6) |
| | Never, 242 (60.2) |
| | Missing, 6 (1.4) |
| Current pain, n (%)* | Experiencing pain, 281 (69.9) |
| | Pain-free, 90 (22.4) |
| | Missing, 31 (7.7) |
| GCPS pain intensity (0–100), mean (SD)** | 36.84 (25.34) |
| GCPS disability score (0–100), mean (SD)** | 30.68 (29.86) |
| GARS total score (18–90), mean (SD)** | 24.75 (11.95) |

(*Continued*)

**Table 1.** (Continued)

| Characteristic | Value |
|---|---|
| UDI (all items) total score (0–80), mean (SD)** | 18.55 (18.41) |

\* Categorised using the 'current' pain intensity item of the Graded Chronic Pain Scale

\*\* Includes imputed missing values

subsequent analyses reported here. In the retained cases, a notable floor effect was observed for several UDI items, with Fig 1 showing that the majority of respondents chose the first option for five out of the eight items. Little's MCAR test, which assesses the randomness of missing data, yielded a chi-squared statistic of 38.44 with 28 degrees of freedom and a $p$-value of 0.09. This suggests the missingness in the retained UDI data was likely random and not influenced by other variables. Further inspection of Fig 1, in which variables are displayed in order of how they appeared in the survey, shows an increasing number of missing values with each successive item. In total, 121 missing UDI values were imputed using K-Nearest Neighbour algorithm. Imputation did not remove the floor effects (see supporting information file for further details).

### Internal consistency

Based on the full sample, including imputed values, the raw Cronbach's α for all eight UDI items was calculated to be 0.92 (95% CI: 0.91, 0.93). By comparison, Cronbach's α for the GARS total sum was 0.97 (95% CI: 0.96, 0.97) and for the three disability (pain interference) numerical rating scale items of the GCPS was 0.95 (95% CI: 0.95, 0.96). For the UDI, Table 2 displays both the item-total correlations and Cronbach's α values if an individual item was dropped.

### Preparation for factor analysis

Bartlett's Test of Sphericity was significant ($\chi^2 = 2115.97$, $df = 28$, $p < 0.001$) and the KMO Measure of Sampling Adequacy was 0.92 for all UDI items, and above 0.90 for each individual UDI item, indicating that the data was suitable for factor analysis. With 364 participants providing UDI data, there were sufficient participants to subset half (182 cases) as a training dataset (providing 22.75 values per UDI item for EFA) and the same number of cases for validation (CFA). A comparison of participant characteristics in the randomly allocated subsets revealed no significant differences (see supporting information file for details). Hence, factor analysis proceeded as planned.

### Number of factors

The results of the four bootstrapped methods used to determine the number of factors to extract are displayed in Fig 2. Taking the modal value of the histograms, parallel analysis, comparative data, MAP, and the number of eigenvalues > 1 all suggested a 1-factor solution. Additionally, the scree plot of bootstrapped eigenvalues (Fig 3) showed that one factor was clearly dominant. Hence, a one-factor model was taken forward for further evaluation.

### Exploratory factor analysis

Given that the EFA model was based on a single factor, Procrustes rotation was not required to align factor loadings from each bootstrap iteration. Table 3 displays the factor loadings on

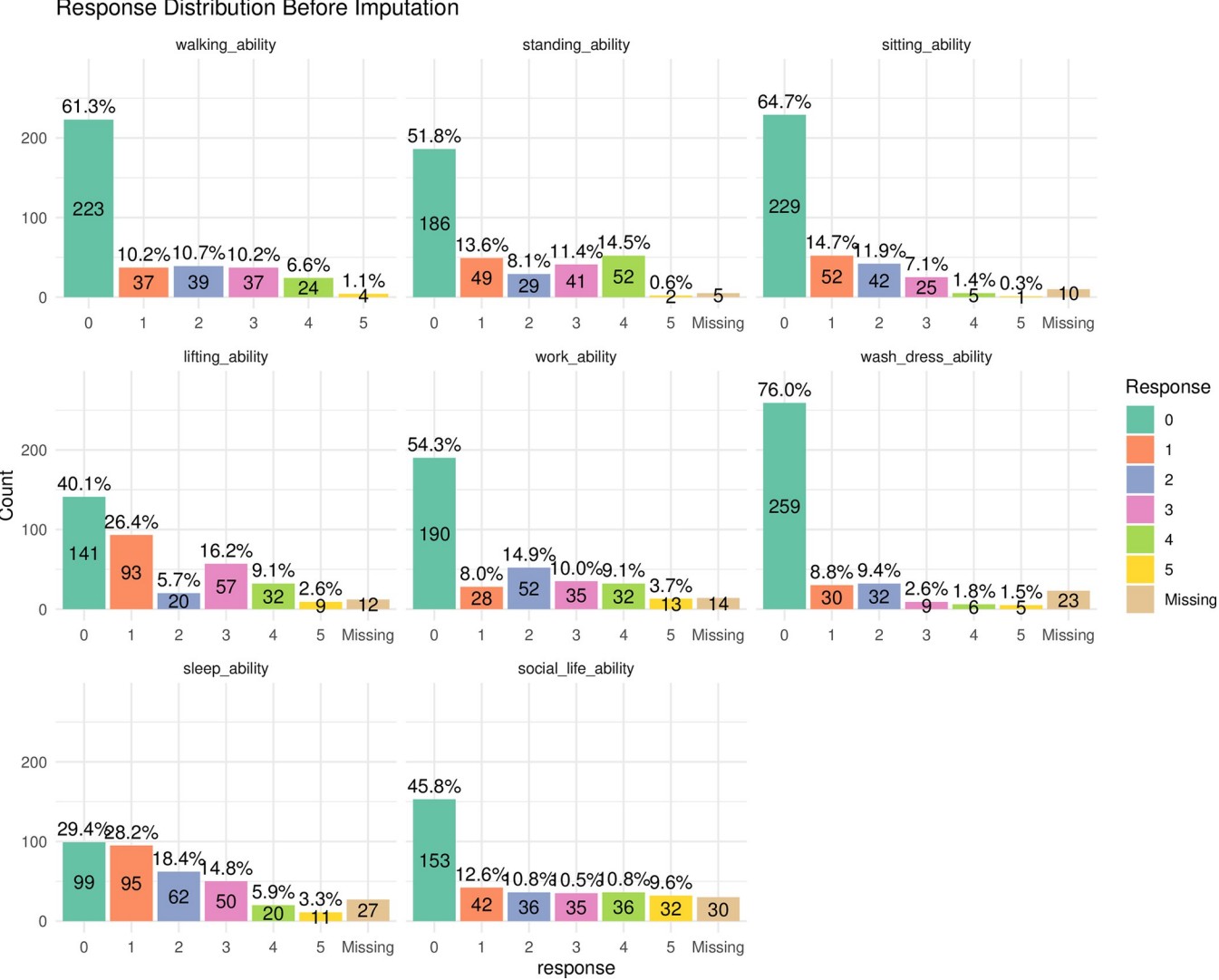

**Fig 1. Response distributions and missing values for UDI items (prior to imputation).**

**Table 2. Internal consistency metrics for all UDI items.**

| UDI item | Item-total correlations | Alpha with item dropped |
|---|---|---|
| *walking* | 0.840 | 0.908 |
| *standing* | 0.868 | 0.905 |
| *sitting* | 0.687 | 0.921 |
| *lifting and carrying* | 0.863 | 0.905 |
| *work and daily routine* | 0.904 | 0.901 |
| *washing and dressing* | 0.764 | 0.916 |
| *sleeping* | 0.659 | 0.925 |
| *social and recreational activities* | 0.855 | 0.909 |

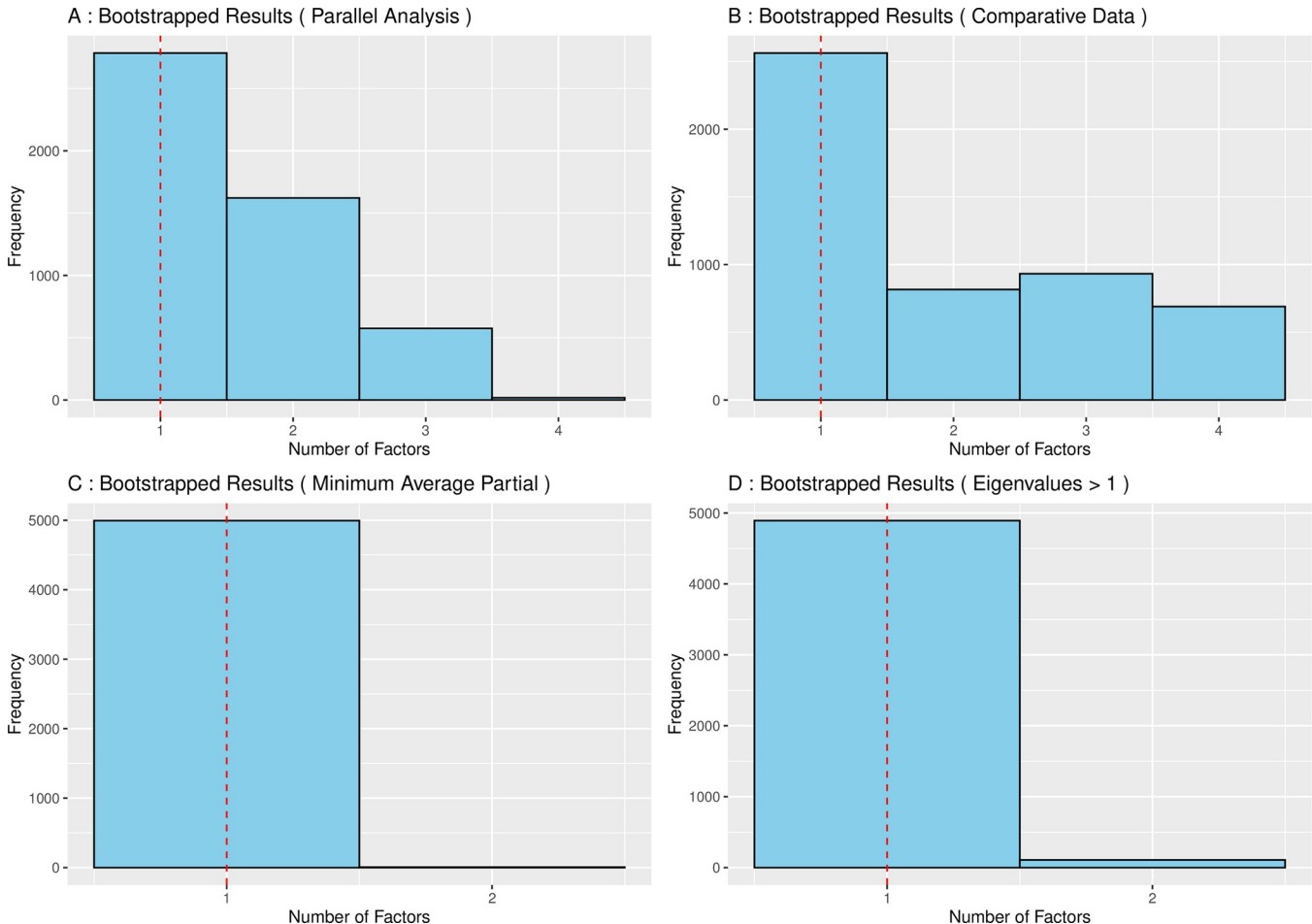

**Fig 2. Number of factors suggested for extraction.**

the training data for all eight UDI items. In this model, the point estimate of all UDI item loadings exceeded the minimum threshold of 0.32 and in no items did the 95% CIs include this threshold value. Therefore, no factor loadings were considered weak.

The sum of squared loadings, reflecting the shared variance among UDI items within the single-factor model, was estimated to be 4.7, with a 95% CI ranging from 4.2 to 5.2. Hence, the single factor captured the variance of approximately five of the eight UDI variables. Accordingly, 59.0% (95% CI: 53.1%, 64.5%) of the total scale variance for the observed UDI items was accounted for, which exceeded the predetermined 50% minimum threshold.

The point estimates of standardised residuals exceeded the 0.5 threshold in only one UDI item (*sitting*). Moreover, the upper bound of the 95% CI for the standardised residuals did not exceed the 0.5 maximum threshold in half of the UDI items, thus meeting predetermined requirements. Inspection of the communalities showed that the lower bound of the 95% CIs exceeded the 0.4 minimum threshold in all but two UDI items (*sitting* and *sleeping*). Indeed, the communality of *sleeping* was notably lower than all other items, while the lower bound of the confidence interval of *sitting* just fell below the 0.4 threshold (Table 3). Hence, the bootstrapped EFA was repeated with these items being sequentially removed (i.e., 7-item and 6-item UDI models were evaluated).

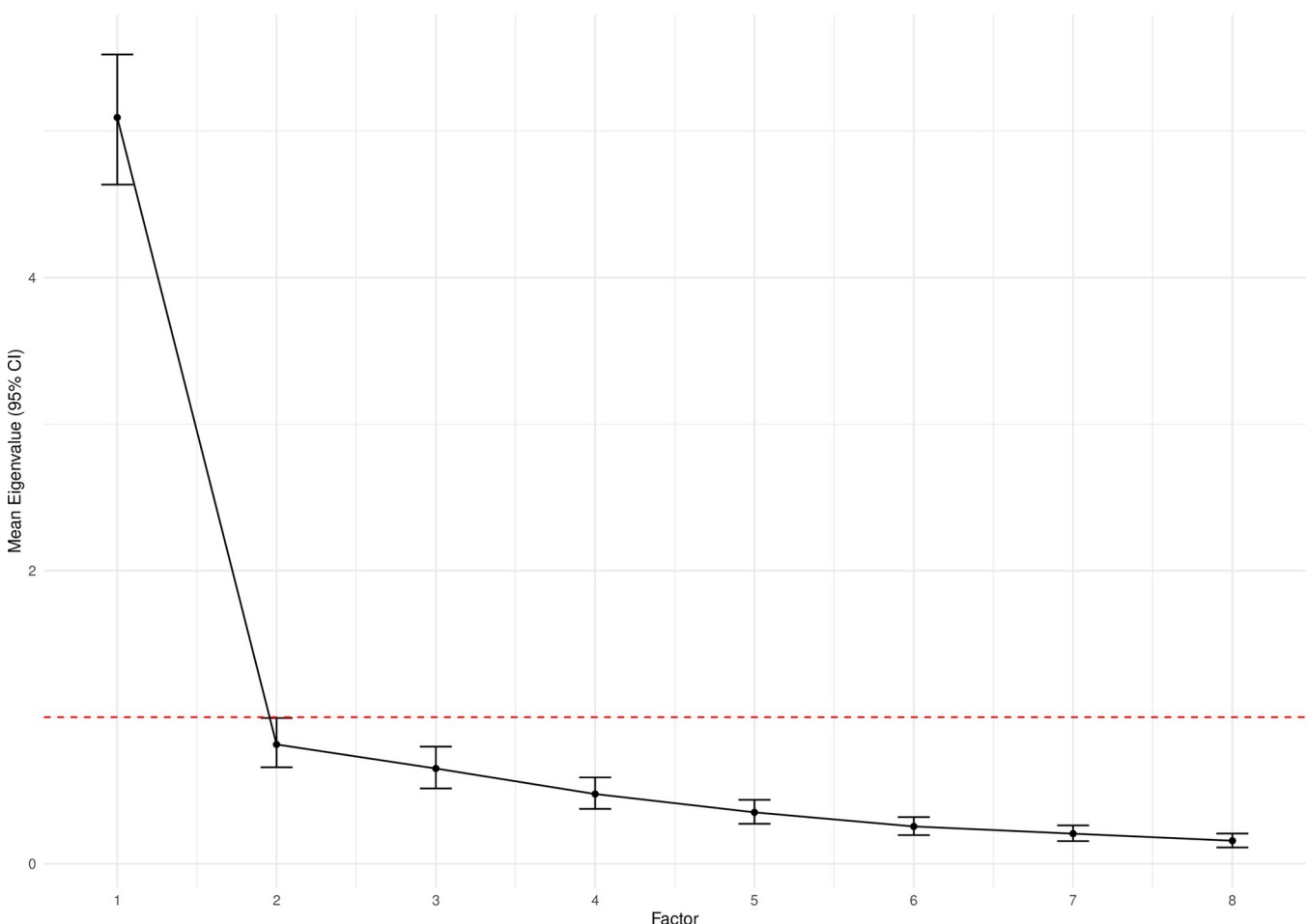

**Fig 3. Scree plot of mean eigenvalues.**

The 182 cases of the training dataset provided 26 values per item for EFA of the 7-item version of the UDI, whereas 30.3 values per item were available for EFA of the 6-item version. A single factor model was advocated by parallel analysis, comparative data, MAP, and the number of eigenvalues > 1 for both 7-item and 6-item EFA models (see supporting information for histograms). Scree plots for these reduced item models (also provided in the supporting information) each showed a single dominant factor. Chronbach's alpha for the 7-item questionnaire was 0.93 (95% CI: 0.92, 0.94) and for the 6-item questionnaire was also 0.93 (95% CI: 0.92, 0.94). The sum of squared loadings for the 7-item model was 4.5 (95% CI: 4.1, 4.9), representing 64.4% (95% CI: 58.2, 70.2) of the observed variance. For the 6-item model, the sum of squared loadings was 4.1 (95% CI: 3.7, 4.4), representing 68.2% (95% CI: 62.2, 73.7) of the variance.

Table 3 enables factor loadings and communalities of these reduced item versions of the UDI to be compared to the original 8-item version. Notably, for items that appear in all three UDI versions, these values were very similar across EFA models. In the 7-item model, the point estimate of standardised residuals exceeded the maximum 0.5 threshold in only one UDI item (*sitting*). However, the upper bound of the 95% CI for these standardised residuals exceeded the 0.5 maximum threshold in more than half of the items, failing to meet predetermined requirements. By contrast, in the 6-item model, not only did the point estimates of standardised residuals not exceed the 0.5 threshold in any items, but also the upper bound of the

**Table 3. Factor loadings, communalities at extraction, and standardised residuals from exploratory factor analysis models.**

| 8-item UDI model | | | |
|---|---|---|---|
| Item | Factor loading | Communality | Residual |
| *work and daily routine* | 0.87 (0.80, 0.93) | 0.77 (0.65, 0.86) | 0.03 (0.02, 0.05) |
| *lifting and carrying* | 0.86 (0.79, 0.92) | 0.74 (0.63, 0.84) | 0.04 (0.02, 0.06) |
| *standing* | 0.82 (0.76, 0.88) | 0.68 (0.57, 0.77) | 0.04 (0.02, 0.05) |
| *social and recreational activities* | 0.81 (0.74, 0.88) | 0.66 (0.55, 0.77) | 0.05 (0.03, 0.08) |
| *walking* | 0.79 (0.71, 0.85) | 0.62 (0.51, 0.72) | 0.03 (0.02, 0.05) |
| *washing and dressing* | 0.73 (0.64, 0.79) | 0.53 (0.41, 0.63) | 0.04 (0.02, 0.05) |
| *sitting* | 0.67 (0.56, 0.77) | 0.46 (0.32, 0.59) | 0.06 (0.04, 0.08) |
| *sleeping* | 0.52 (0.38, 0.64) | 0.27 (0.14, 0.41) | 0.04 (0.02, 0.08) |
| 7-item UDI model | | | |
| Item | Factor loading | Communality | Residual |
| *work and daily routine* | 0.87 (0.80, 0.92) | 0.76 (0.63, 0.85) | 0.03 (0.02, 0.05) |
| *lifting and carrying* | 0.87 (0.80, 0.92) | 0.75 (0.65, 0.84) | 0.04 (0.02, 0.06) |
| *standing* | 0.83 (0.76, 0.89) | 0.69 (0.58, 0.79) | 0.03 (0.02, 0.05) |
| *social and recreational activities* | 0.81 (0.73, 0.88) | 0.65 (0.53, 0.77) | 0.04 (0.02, 0.06) |
| *walking* | 0.81 (0.73, 0.87) | 0.65 (0.54, 0.76) | 0.04 (0.02, 0.06) |
| *washing and dressing* | 0.77 (0.70, 0.83) | 0.60 (0.49, 0.69) | 0.03 (0.02, 0.05) |
| *sitting* | 0.64 (0.52, 0.75) | 0.41 (0.27, 0.56) | 0.06 (0.04, 0.09) |
| 6-item UDI model | | | |
| Item | Factor loading | Communality | Residual |
| *work and daily routine* | 0.87 (0.81, 0.93) | 0.76 (0.65, 0.86) | 0.03 (0.01, 0.05) |
| *lifting and carrying* | 0.85 (0.78, 0.90) | 0.72 (0.61, 0.82) | 0.02 (0.01, 0.04) |
| *standing* | 0.83 (0.76, 0.89) | 0.69 (0.58, 0.79) | 0.03 (0.02, 0.05) |
| *social and recreational activities* | 0.83 (0.76, 0.89) | 0.67 (0.54, 0.79) | 0.03 (0.02, 0.05) |
| *walking* | 0.82 (0.74, 0.89) | 0.69 (0.58, 0.78) | 0.03 (0.02, 0.04) |
| *washing and dressing* | 0.75 (0.67, 0.81) | 0.56 (0.46, 0.66) | 0.03 (0.01, 0.04) |

95% CIs did not exceed this threshold for any item. The three competing single-factor UDI models were taken forward to be evaluated further using CFA.

## Confirmatory factor analysis

Fig 4 presents the CFA path diagrams for 8-item, 7-item, and 6-item UDI models respectively, with path coefficients being loadings onto the single factor relative to the loading of the '*work and daily routine*' item. CFA model fit indices for the three single-factor models (8-items, 7-items, and 6-items) are displayed in Table 4. With the exception of RMSEA, the point estimates (means) of all fit indices met their predetermined acceptable thresholds in every model (chi-square test $p > 0.05$; CFI > 0.90; TLI > 0.90; RMSEA < 0.08; and SRMR < 0.05). The mean and upper 95% CI of RMSEA exceeded the predetermined threshold value of 0.08 in every model. However, the lower bound of the RMSEA confidence interval met or was below this threshold in the 8-item and 6-item models. Notably, the AIC and BIC values reduced with each successive item removal (*sleeping* followed by *sitting*) and their 95% CIs did not overlap, indicating statistically significant differences between the three models.

## External validation of the UDI

Correlations between the various disability measures, including the 8-item, 7-item, and 6-item versions of the UDI are displayed in Table 5. Notably, the point estimates (means) of all three

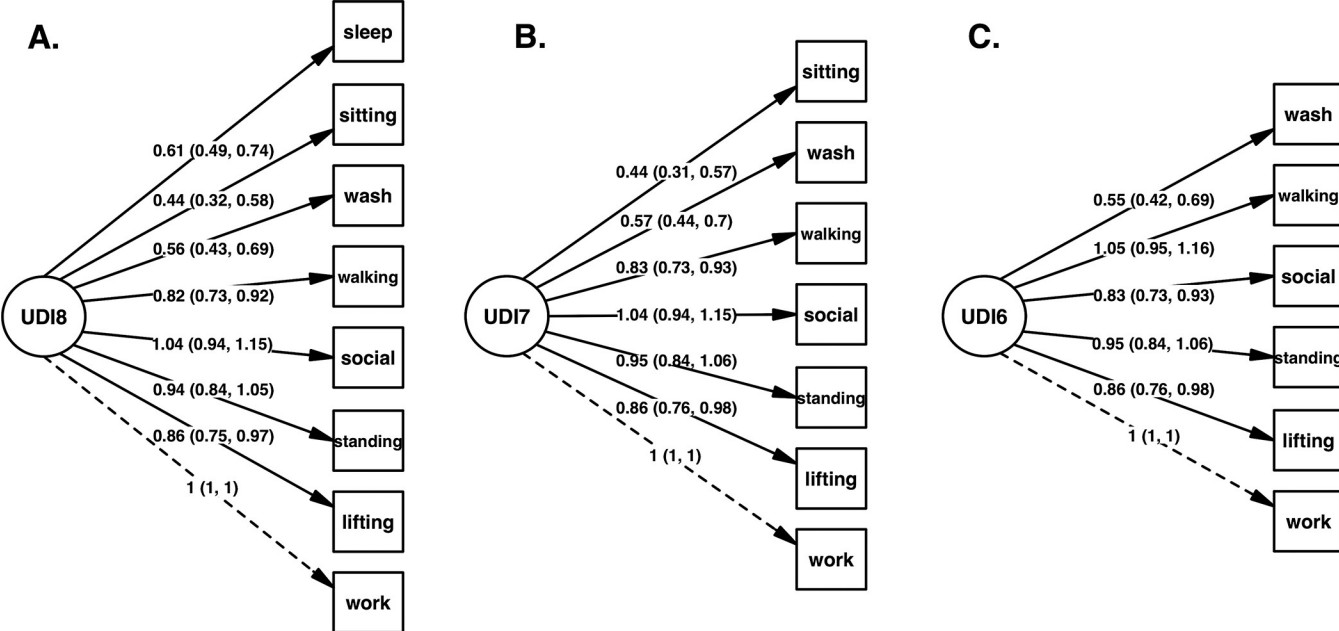

**Fig 4. Confirmatory factor analysis (CFA) models for UDI items.** Path coefficients are factor loadings relative to the '*work and daily routine*' item. A. 8-item model. B. 7-item model. C. 6-item model.

UDI versions demonstrated very similar, strong ($r > 0.7$) correlations with the GARS and the GCPS disability scores. By comparison, at $r = 0.63$ (95% CI: 0.56, 0.70), the correlation between the GARS and GCPS disability did not reach this threshold. Additionally, only the GCPS disability score was strongly correlated with the GCPS pain intensity score ($r = 0.84$ [95% CI: 0.808, 0.876]). In the subset of participants with non-zero current pain and at least one completed UDI item (n = 276), point estimates (means) of these correlations were very similar to those of the full dataset: the UDI8 was strongly correlated with both the GARS ($r = 0.812$ [95% CI: 0.767, 0.850]) and the GCPS disability ($r = 0.713$ [95% CI: 0.640, 0.778]). Full details of this

**Table 4. Fit indices for confirmatory analysis models (mean, 95% CI).**

| Fit measure | 8-item UDI model | 7-item UDI model | 6-item UDI model |
|---|---|---|---|
| Chi-square | 77.76 (45.12, 121.28) | 64.25 (34.06, 105.24) | 36.12 (14.16, 66.98) |
| | df = 20, $p = 0.96$ | df = 14, $p = 0.91$ | df = 9, $p = 0.84$ |
| CFI | 0.95 (0.91, 0.98) | 0.95 (0.92, 0.98) | 0.97 (0.94, 0.99) |
| TLI | 0.93 (0.88, 0.97) | 0.93 (0.87, 0.97) | 0.95 (0.90, 0.99) |
| RMSEA | 0.12 (0.08, 0.17) | 0.14 (0.09, 0.19) | 0.12 (0.06, 0.19) |
| SRMR | 0.04 (0.03, 0.05) | 0.04 (0.02, 0.05) | 0.02 (0.01, 0.03) |
| AIC | 3998 (3801.39, 4182.04) | 3451 (3257.40, 3634.72) | 2989 (2820.14, 3145.41) |
| BIC | 4075 (3878.29, 4258.94) | 3518 (3324.69, 3702.00) | 3047 (2877.81, 3203.08) |

CFI = Comparative fit index

TLI = Tucker-Lewis index

RMSEA = Root mean square error of approximation

SRMR = Standardised root mean square residual

AIC = Akaike Information Criterion

BIC = Bayesian Information Criterion

**Table 5. Correlation matrix of disability measures.**

|  | UDI 8 total | UDI 7 total | UDI 6 total | GARS total | GCPS disability |
|---|---|---|---|---|---|
| UDI 7 total | 0.993 *** (0.991, 0.994) |  |  |  |  |
| UDI 6 total | 0.987 *** (0.985, 0.990) | 0.995 *** (0.994, 0.996) |  |  |  |
| GARS total | 0.827 *** (0.791, 0.859) | 0.840 *** (0.807, 0.869) | 0.834 *** (0.800, 0.864) |  |  |
| GCPS disability | 0.735 *** (0.673, 0.791) | 0.722 *** (0.655, 0.782) | 0.719 *** (0.651, 0.779) | 0.631 *** (0.556, 0.702) |  |
| GCPS pain intensity | 0.611 *** (0.543, 0.677) | 0.593 *** (0.519, 0.659) | 0.586 *** (0.510, 0.652) | 0.500 *** (0.419, 0.580) | 0.844 *** (0.808, 0.876) |

\* $p < 0.05$

\*\* $p < 0.01$

\*\*\* $p < 0.001$

sensitivity analysis are provided in the supporting information. All correlations (in both the full dataset and subset) were statistically significant with $p < 0.001$.

Table 6 shows clear differences between lower and upper quartiles of each version of the UDI and all comparator variables: the GARS total score, GCPS disability score, and GCPS pain intensity score. The Mann-Whitney U tests confirmed that these differences were significant in all variables. These quartiles could therefore be used to evaluate the discriminatory ability of the three UDI versions.

The results from the bootstrapped ROC curve analyses (Table 7) showed that all three UDI versions performed very well in discriminating between upper and lower quartiles of these comparator variables. All AUC mean values were close to 1, indicating excellent model performance for all comparisons. The mean values for both sensitivity and specificity were high, which shows good discriminatory ability. The optimal threshold scores varied for different comparisons, which was expected as these depend on the distribution of the predictor variable in each comparison. The 95% CIs of these estimates were very narrow, suggesting that estimates were very stable across bootstrapped samples.

## Discussion

The development and initial testing of a brief, generic self-reported disability questionnaire, the Universal Disability Index (UDI), has been described through a rigorous methodological process on a sufficiently large general population sample. Comparisons with well-established existing disability questionnaires provided external validation. All tested versions of the UDI (8-item, 7-item, and 6-item) performed well at every stage of assessment.

Efforts were made to ensure methodological rigour during the assessment process. Specifically, bootstrap resampling was used whenever possible to ensure enhanced robustness and interpretability of statistical estimates, including factor loadings, error variances and model fit indices. This technique provides insights into the stability and generalisability of the results

**Table 6. Differences between lower and upper quartiles of scales.**

| Variable (range) | Lower quartile score | Lower quartile mean (SD) | Upper quartile score | Upper quartile mean (SD) | Mann-Whitney U $p$-value |
|---|---|---|---|---|---|
| UDI 8 total (0–80) | 2 | 1.0 (1.0) | 32 | 45.6 (9.9) | 1.82E-34 |
| UDI 7 total (0–70) | 1.5 | 0 (0) | 28 | 39.9 (9.3) | 7.43E-36 |
| UDI 6 total (0–60) | 0 | 0 (0) | 26 | 36.6 (8.2) | 1.16E-36 |
| GARS total (18–72) | 18 | 18 (0) | 27.3 | 41.7 (13.1) | 6.16E-62 |
| GCPS disability (0–100) | 3.3 | 0.7 (1.3) | 53.3 | 74.9 (14.3) | 3.10E-35 |
| GCPS pain intensity (0–100) | 16.7 | 7.5 (6.2) | 53.3 | 69.1 (12.6) | 4.90E-36 |

**Table 7. ROC curve scores for UDI versions.**

| Comparator | AUC, mean (95% CI) | Sensitivity, mean (95% CI) | Specificity, mean (95% CI) | Optimal threshold, mean (95% CI) |
|---|---|---|---|---|
| *UDI 8 total* | | | | |
| GARS total | 0.9788 (0.9785, 0.9791) | 0.9527 (0.9519, 0.9535) | 0.9444 (0.9436, 0.9452) | 21.8322 (21.7349, 21.9295) |
| GCPS disability | 0.9601 (0.9597, 0.9605) | 0.9590 (0.9583, 0.9598) | 0.9045 (0.9036, 0.9053) | 14.4372 (14.3869, 14.4875) |
| GCPS pain intensity | 0.9174 (0.9169, 0.9180) | 0.8950 (0.8939, 0.8961) | 0.8606 (0.8594, 0.8618) | 11.5896 (11.5111, 11.6681) |
| *UDI 7 total* | | | | |
| GARS total | 0.9804 (0.9801, 0.9806) | 0.9410 (0.9403, 0.9417) | 0.9629 (0.9624, 0.9634) | 19.4664 (19.4046, 19.5282) |
| GCPS disability | 0.9545 (0.9541, 0.9550) | 0.9634 (0.9627, 0.9641) | 0.8879 (0.8869, 0.8888) | 9.0246 (8.9848, 9.0644) |
| GCPS pain intensity | 0.9126 (0.9120, 0.9132) | 0.8893 (0.8882, 0.8904) | 0.8563 (0.8551, 0.8575) | 7.4464 (7.3945, 7.4983) |
| *UDI 6 total* | | | | |
| GARS total | 0.9794 (0.9792, 0.9797) | 0.9328 (0.9320, 0.9337) | 0.9632 (0.9624, 0.9639) | 18.3246 (18.2502, 18.3990) |
| GCPS disability | 0.9514 (0.9509, 0.9519) | 0.9572 (0.9565, 0.9579) | 0.8802 (0.8792, 0.8812) | 7.7914 (7.7238, 7.8590) |
| GCPS pain intensity | 0.9068 (0.9062, 0.9074) | 0.8802 (0.8789, 0.8815) | 0.8503 (0.8489, 0.8517) | 7.1118 (7.0336, 7.1900) |

AUC = area under the curve

across different population subsets [54, 55]. It also facilitated the empirical testing of data distribution assumptions, reducing potential bias and contributing to the overall validity of the study's findings. In addition, bootstrapping was combined with a triangulation of four established methods to identify the correct number of factors to extract, which was a single factor even when items were removed. Hence, confidence in the factor structure can be high.

During the EFA modelling, all UDI items loaded strongly onto the single factor in all versions that were tested, even with the imposition of a strict criterion that the 95% CIs did not include a predetermined minimum value of 0.32 or below. Standardised residuals of UDI items, which represent the differences between the observed and expected covariance matrices, were also reassuringly low, suggesting a good EFA model fit. However, two UDI items (*sleeping* and *sitting*) had to be considered for removal because the 95% CIs of their communalities included values below the predetermined 0.4 minimum threshold. A communality of less than 0.4 means that the variable shares relatively little common variance with the other variables in the factor analysis [56]. It is perhaps unsurprising that the communalities of these two items were lower than other UDI items since both represent more 'static' ADLs than the others embodied within the scale [60].

All tested versions of the UDI (8-item, 7-item, and 6-item) performed well during CFA modelling. With the exception of RMSEA, which quantifies how well the model approximates the population covariance matrix, all fit indices met predetermined thresholds that indicate a good fit to the data. The RMSEA point estimate of the 8-item and 6-item UDI model was 0.12, while the 7-item model reached 0.14, all well above the target threshold of 0.08. Given its sensitivity to model misspecification and sample size, RMSEA should be closely monitored in future studies evaluating the UDI, particularly those involving different population subsets or methodological approaches. The respective AIC and BIC values reduced with each successive item removal suggesting that the 6-item model fitted the data best.

One advantage of fewer items in a self-reported disability questionnaire is a lower burden for participants. This makes the 6-item version of the UDI a good candidate for use in higher frequency data collection, such as longitudinal studies requiring regular measurements of disability for months or even years [61–64]. Given a desire for both parsimony and brevity, it is therefore difficult not to fully endorse the 6-item version of the UDI. On the other hand, both *sitting* and *sleeping* items may offer potentially important theoretical contributions when

measuring an individual's disability. *Sitting* captures the primary occupational behaviour of a large proportion of the population. Moreover, people who are confined to sitting (i.e., wheel-chair users) would have a marked reduction in participation if they also lost the ability to sit for long periods. By contrast, sleeping is a universal behaviour that everybody must perform; a necessity regardless of age, employment status, predilection for an active or sedentary lifestyle, and is known to be essential for good health. Indeed, if a person's lifestyle was very sedentary and did not involve much walking, standing, lifting, work or social life, they would surely still need to sit and sleep. It is therefore plausible that the high prevalence of low disability levels in the current sample may not have provided a sufficient test of the potential virtues of incorpo-rating *sleeping* and *sitting*. Given these considerations, both 8-item and 6-item versions of the UDI ought to be retained pending further evaluation within specific clinical populations.

All versions of the UDI correlated strongly (achieved a point estimate of $r > 0.7$) with the two comparator disability measures, the GARS [19] and the GCPS [41, 42] disability score. This suggests that the UDI shares a common construct with each of these measures. Interest-ingly, the correlation between the GARS and GCPS disability was not as strong ($r = 0.63$ [95% CI: 0.56, 0.70]), although this could be partly explained by the different recall periods of the GCPS disability (4-weeks) and GARS (current time). Furthermore, only the disability score of the GCPS was strongly correlated with its pain intensity score ($r = 0.84$ [95% CI: 0.81, 0.88]), indicating construct overlap between these two scores and suggesting that the GCPS concep-tion of disability may be rather narrow. The similarity in magnitudes of correlation coefficients for the full dataset with the subset of participants with non-zero current pain supports the validity of the UDI in capturing important aspects of disability, irrespective of the potential origins of this disability. It also indicates that the statistical models from which estimates were gained in this study are likely to be stable. Taken together, these results confirm that the GARS (which measures independence in the performance of ADLs) and GCPS (which measures pain interference during ADLs) capture different dimensions of disability, while the UDI appears to capture elements common to both GARS and GCPS, yet more than each in isolation. ROC curve analyses of upper and lower quartiles of the comparator disability scales showed that every version of the UDI possessed excellent discriminatory ability, with each AUC, sensitivity and specificity, and their 95% CIs, well above 0.90 in every case. The equivalent values for GCPS pain intensity were slightly lower, which is in line with expectations since this is not a measure of disability. Based on these comparisons, all versions of the UDI were shown to pos-sess concurrent and discriminant validity.

The internal consistency of the UDI items was also excellent: the raw Cronbach's α for all eight items was calculated to be 0.92 (95% CI: 0.91, 0.93), with *sleeping* removed was 0.93 (95% CI: 0.92, 0.94), and with both *sleeping* and *sitting* removed was also 0.93 (95% CI: 0.92, 0.94). This compares well against the comparator disability scales: 0.97 (95% CI: 0.96, 0.97) for the GARS and 0.95 (95% CI: 0.95, 0.96) for the GCPS disability items.

The good performance of the UDI items across the various tests performed in this study is perhaps to be expected given that the ADLs selected for the UDI, and the wording and ordinal steps of their response items, were drawn from highly successful, well-established disability questionnaires, the ODI [43, 44] and the NDI [45] (the latter being developed from the for-mer). Credit is therefore due to the developers of these existing questionnaires and those who have tested them extensively over the years. However, the UDI is different from its pain-focused parent questionnaires in a very important way: it is entirely devoid of any reference or attribution to any disease, condition, or cause of disability.

There are several sub-categories of disability that can be measured, and arguably the most commonly assessed is pain-related disability. The traditional approach to framing pain-related disability questions [41, 43, 65] results in multi-dimensional questions, such as '*How much has*

*your pain interfered with your ability to walk*?' Such questions incorporate two independent constructs—*pain* and *walking* in this example—that the respondent must simultaneously consider. This multidimensionality can conflate the constituent constructs, increasing the challenge for the respondent to simultaneously process multiple pieces of information and then integrate them into a single response. In this example, the respondent must evaluate their ability to walk *through the lens of their experience of pain*. An obvious scenario in which potentially important information would be lost here is if a respondent has a significant reduced ability to walk that they do not attribute to pain (e.g., a congenital or neurological issue). Information about their actual walking ability is not provided, being effectively 'filtered' by the requirement to consider only the aspects of walking that relate to pain. While some might argue that this 'filtering' effect is useful, researchers and clinicians must be mindful that the required attribution of reduced abilities (to pain in this example) must rely entirely on the respondent's own interpretation. This becomes problematic when considering the known complexity of pain and what the respondent should considered to be 'pain-related', which may be very different from the opinion of the clinician or the researcher. Indeed, the past three decades has seen growing evidence to support the notion that a person's psychosocial status such as learned beliefs [66], fears [67], and perceptions [68] can influence the magnitude of their disability. Importantly, respondents are often likely unaware of the relationships between these constructs and their own perceived activity limitations and participation restrictions, and certainly may not attribute them directly to their pain. One established model with strong face validity is the fear-avoidance model [69–72]. When asking a respondent with established pain-related fear-avoidance to consider the origin of their perceived reluctance to perform an ADL such as walking or lifting, they may not attribute their limitations directly to pain. Yet, previously experienced pain may have initiated the development of their fear-avoidance, which might have subsequently become entrenched and manifest as disabling behavioural patterns or even habits. Consequently, their disability may still legitimately be regarded as pain-related, even if it is not explicitly attributed to pain by the respondent.

## Limitations

As with any self-reported questionnaire, the UDI requires that the respondent possess sufficient mental capacity to understand and respond to the questions. If this criterion is unmet, disability can only be assessed using third party observation, with instruments such as the Katz Index of Independence in Activities of Daily Living [7] or the Barthel Index [8]. On the other hand, third-party observations of ADLs are difficult to perform remotely, and currently require an observer who is trained to be familiar with the ADLs being assessed.

The convenience sampling utilised in this study successfully recruited a sufficiently large sample to divide the dataset into training (EFA) and validation (CFA) subsets. The sample was mostly female and white, which is fairly typical of health surveys based in the United Kingdom [68, 73, 74]. Currently, the UDI has only been tested in the English language. Hence, further studies will be required to create and test any translations of the UDI. Likewise, those with visual impairments would need the questions to be presented via a different medium (e.g., audio or braille) to provide a response. Hence, future work should look at different formats and facilitate the recruitment of under-represented groups to ensure the UDI items are appropriate and meaningful in these groups.

Effort was made to recruit individuals with a range of health conditions, and this was achieved with some success. The sample recruited in this study therefore consisted of a mixture of both healthy participants and individuals with varying levels of disability. This sampling framework was deliberate and intended to test the ability of the UDI to discriminate between

high and low severity disability. Even so, a potential weakness of this study is that a relatively low average disability level was seen in both the mean GARS score (24.75 [SD: 11.95] out of a possible 90) and mean GCPS disability score (30.68 [SD: 29.8] from a possible 100), which was likely reflected in the floor effect that was seen in the majority of UDI items in the current dataset. This floor effect could have implications for the analyses, potentially leading to underestimated factor loadings in EFA and CFA, reduced model fit indices, and weaker correlations with comparator questionnaires. One potential source of this floor effect may have been inherited from the ordinal response options drawn from the ODI, which are known to result in floor effects in samples with moderate levels of disability [75]. Yet, the ability to distinguish between more severe levels of disability is also valuable. In future studies, this will need to be addressed by testing the UDI with individuals with a higher prevalence of more severe disability.

There was evidence of questionnaire fatigue amongst respondents with an increasing number of missing values with each successive UDI item. The resulting fewer datapoints for later items will have introduced some response bias. In future studies, this bias can be removed by the use of random item ordering, which is relatively straightforward to implement in online surveys.

Finally, data were collected from each participant at only one session. Hence, no examination of test-retest reliability is possible with the current dataset. Additional analyses (e.g., sensitivity to chance, minimally important clinical difference, etc.) should be investigated in future studies before the UDI is routinely used in a clinical setting. Reliability studies should be a priority for future work.

## Conclusions

The Universal Disability Index (UDI) is a brief, generic self-reported disability questionnaire that appears to be valid and to possess good psychometric properties. The UDI has a single factor structure and either a 6-item, 7-item or 8-item version can be used to measure disability. A desire for parsimony and brevity suggests that the 6-item version of the UDI should be recommended but further testing of all versions is warranted in clinical populations to help support this decision.

## Supporting information

**S1 Table. Wording of UDI questions and response options.**
(PDF)

**S2 Table. Comparisons of categorical variables between EFA and CFA datasets.**
(PDF)

**S3 Table. Test statistics for comparisons of categorical variables between EFA and CFA datasets.**
(PDF)

**S4 Table. Comparisons of continuous variables between EFA and CFA datasets.**
(PDF)

**S5 Table. Test statistics for comparisons of continuous variables between EFA and CFA datasets.**
(PDF)

**S6 Table. Correlation matrix of disability measures for subset of participants with non-zero current pain intensity.**
(PDF)

**S1 Fig. Response distributions for UDI items after imputation.**
(TIF)

**S2 Fig. Number of factors suggested for extraction (7-item UDI model).**
(TIF)

**S3 Fig. Scree plot of mean eigenvalues (7-item UDI model).**
(TIF)

**S4 Fig. Number of factors suggested for extraction (6-item UDI model).**
(TIF)

**S5 Fig. Scree plot of mean eigenvalues (6-item UDI model).**
(TIF)

## Acknowledgments

The author would like to thank Mitchell Scanlan and Shannon Munks for their assistance with distributing the public survey invitation, and Nadège Haouidji-Javaux for input with the RED-Cap project.

## Author Contributions

**Conceptualization:** David William Evans.

**Data curation:** David William Evans.

**Formal analysis:** David William Evans.

**Investigation:** David William Evans.

**Methodology:** David William Evans.

**Project administration:** David William Evans.

**Resources:** David William Evans.

**Software:** David William Evans.

**Supervision:** David William Evans.

**Validation:** David William Evans.

**Visualization:** David William Evans.

**Writing – original draft:** David William Evans.

**Writing – review & editing:** David William Evans.

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
