## [Decision Letter · Decision Letter 0]

23 Oct 2023

PONE-D-23-31316Development and initial testing of a brief, generic self-reported disability questionnaire: the Universal Disability IndexPLOS ONE

Dear Dr. Evans,

Thank you for submitting your manuscript to PLOS ONE. After careful consideration, we feel that it has merit but does not fully meet PLOS ONE’s publication criteria as it currently stands. Therefore, we invite you to submit a revised version of the manuscript that addresses the points raised during the review process.

We look forward to receiving your revised manuscript.

Kind regards,

Mohammad Asghari Jafarabadi

Academic Editor

PLOS ONE

Journal Requirements:

Reviewers' comments:

Reviewer's Responses to Questions

**Comments to the Author**

1. Is the manuscript technically sound, and do the data support the conclusions?

Reviewer #1: Yes

Reviewer #2: Yes

2. Has the statistical analysis been performed appropriately and rigorously? 

Reviewer #1: Yes

Reviewer #2: Yes

3. Have the authors made all data underlying the findings in their manuscript fully available?

Reviewer #1: Yes

Reviewer #2: Yes

4. Is the manuscript presented in an intelligible fashion and written in standard English?

Reviewer #1: Yes

Reviewer #2: Yes

5. Review Comments to the Author

Reviewer #1: Congratulations to the authors for a very well conducted and adequately reported study; however, there are some scopes of modification and improving the manuscript.

1. Introduction is divided into too many paragraphs. Kindly merge those meaningfully into lesser number of paragraphs.

2. How was the sample size determined? Please elaborate.

3. Kindly provide the diagram of the CFA model.

4. Please rephrase the 1st sentence of conclusion.

Reviewer #2: This was a very well-written manuscript and the statistical analysis for validation of the UDI was reported meticulously. The authors have provided reasonable justifications for development of a universal tool to measure disability across different conditions. There are some queries and suggestions that I hope the authors can address.

1. Bearing in mind that there are various types of disability, which may include physical, mental, emotional and social disability. How well can the UDI be relevant to different types of disability such as those due to mental or emotional illnesses? It may be necessary to revise that UDI is aimed at measuring universal physical disability rather than all types of disability.

1. The UDI items were derived from the Oswestry Disability Index and Neck Disability Index which were validated to measure disability due to pain. The item for pain was not included into the UDI. Was there a reason for this?

2. There are other established generic disability tools such as Modified Barthel Index , or Katz Index of Independence in Activities of Daily Living. These are used in various types of conditions as well. How does the UDI compare with these tools? Thus far, the authors have discussed the comparison of UDI with GARS and very few other tools.

3. The UDI was compared against GARS and GCPS. Comparison with GARS is suitable. However, I would like to know why the authors chose a pain-related scale for comparison. While pain is prevalent, not all disabilities are associated with pain. Patients with neurological disorders with loss of sensation may not experience pain but they still have disability. Therefore, did the authors select out participants who had pain for the analysis which compared the UDI and GCPS? Otherwise, those without pain might have discrepantly lower scores for GCPS. This could have affected the correlation between UDI and GCPS pain intensity. Besides presenting the mean scores for their pain intensity in Table 1, do present the categorical results as well. e.g. How many n(%) had no pain, mild pain, moderate pain etc.

4. Table 1, I noticed that educational attainment was presented as age of having left formal education. This is probably personal preference but why didn't the authors present it according to levels of formal education rather than on age of leaving full time education? This is however a minor comment.

5. Some results which have been presented in the tables do not need to be repeated in the main text, as the manuscript is already very long and wordy.

6. Table 7 displays numbers with 9 decimal places. Perhaps this can be converted into max 3 decimal points.

I hope these comments are helpful for the authors to improve on the manuscript. Good job!

6. PLOS authors have the option to publish the peer review history of their article (what does this mean?). If published, this will include your full peer review and any attached files.

Reviewer #1: **Yes: **Subhranil Saha

Reviewer #2: No

---

## [Author Response · Author response to Decision Letter 0]

24 Oct 2023

Reviewer #1: Congratulations to the authors for a very well conducted and adequately reported study; however, there are some scopes of modification and improving the manuscript.

1. Introduction is divided into too many paragraphs. Kindly merge those meaningfully into lesser number of paragraphs.

Thank you for this suggestion. Having tried to merge paragraphs of the introduction in different ways, it has difficult to make this work. The first two paragraphs were merged, but upon re-reading them, this merger does not work well. Based on the following guidance, we feel the original paragraph divisions probably should be retained.

When to start a new paragraph (https://libguides.newcastle.edu.au/writing-paragraphs/structure):

• Start new main points or new ideas in a new paragraph. If you have an extended idea across multiple paragraphs, each new point within that idea should have its own paragraph.

• Use a new paragraph to introduce a contrasting or different position. Use a clear topic sentence to identify the main idea. 

• If the paragraph becomes too long or the material is overly complex, you will need to create a break to make your writing more readable. Try splitting long paragraphs into two shorter paragraphs. This means you will need to write a new topic sentence at the start of the new paragraph.

• Introductions and conclusions are usually written as separate paragraphs.

2. How was the sample size determined? Please elaborate.

Factor analysis was the primary analysis of the study that determined sample size requirements. Sample sizes for factor analysis are unfortunately not straightforward but, in general, greater numbers do provide increased accuracy of estimates, which is why we used a minimum target sample size and did not specify an upper limit. The minimum target sample size in this study to perform both exploratory factor analysis (EFA) and confirmatory factor analysis (CFA) (320 respondents) was based on stated requirements for each stage (minimum of 20 responses on each of the 8 UDI items). This was explicitly mentioned on page 12 of the original manuscript:

Depending on the size of the available sample, at least 20 cases per UDI item [1] (i.e., at least 160 cases) were to be selected for a training dataset for exploratory factor analysis (EFA). The number was also anticipated to be necessary for a validation dataset to perform confirmatory factor analysis (CFA) [2]. If the study sample was sufficiently large (i.e., 320 or more useable responses), the sample would be randomly divided into training and validation subsets; otherwise, the full dataset would be used only for EFA, and CFA would not be performed. In the event of data sub-setting, the distribution of participant characteristics between the two subsets would be compared.

And in results, it was stated (page 20):

With 364 participants providing UDI data, there were sufficient participants to subset half (182 cases) as a training dataset (providing 22.75 values per UDI item for EFA) and the same number of cases for validation (CFA).

3. Kindly provide the diagram of the CFA model.

Thank you for suggesting this. The path diagrams for the 8-item, 7-item and 6-item CFA models have now been added as an additional figure (Fig 4. a, b, c).

4. Please rephrase the 1st sentence of conclusion.

Thank you for pointing out this opportunity to improve the manuscript. The original first sentence of the conclusion was:

“Through an evaluation using rigorous methodology, a brief, generic self-reported disability questionnaire was found to be valid and possess good psychometric properties.”

This has now been rephrased to:

“The Universal Disability Index (UDI) is a brief, generic self-reported disability questionnaire that appears to be valid and to possess good psychometric properties.”

Many thanks indeed to Reviewer 1 for spending their valuable time in providing feedback on this work.

Reviewer #2: 

This was a very well-written manuscript and the statistical analysis for validation of the UDI was reported meticulously. The authors have provided reasonable justifications for development of a universal tool to measure disability across different conditions. There are some queries and suggestions that I hope the authors can address.

Many thanks for these positive comments.

1. Bearing in mind that there are various types of disability, which may include physical, mental, emotional and social disability. How well can the UDI be relevant to different types of disability such as those due to mental or emotional illnesses? It may be necessary to revise that UDI is aimed at measuring universal physical disability rather than all types of disability.

Thank you for this thoughtful comment. This issue was considered in depth so Reviewer 2’s comment is appreciated.

One of the major reasons for deliberately including broad ADL categories such as ‘work and daily routine', ‘social and recreational activities’ and ‘sleeping’ in the UDI was to create a measure of disability caused by physical impairments and/or mental health problems, including emotional and behavioural factors. Note that the question phrasing of the UDI did not follow that of the ODI or NDI, instead taking the form of: “How much have you been able to …”, and therefore being deliberately wide-ranging and much more inclusive than these other questionnaires. Hence, an entirely unattributed, generic measure such as the UDI should be able to capture disability manifesting from non-physical causes. Indeed, there are several examples where mental health can manifest as interference with activities of daily living. For example, when measured using disability-adjusted life-years (DALYS) (the sum of years lived with disability and years of life lost), mental health disorders have remained among the top ten leading causes of burden worldwide for the past 30 years [3].

In this sense, the UDI reflects the approach of the World Health Organization (embodied within the ICF) as being neutral to aetiology, instead placing the emphasis on function rather than a specific condition or disease. In the ICF, disability refers to functional or structural impairments, limitations in personal activities, and restrictions on social participation within the context of environmental factors. In other words, disability denotes the negative outcomes of the interaction between an individual's health condition and contextual factors (environmental and personal factors) that may affect their health.

In the current study, diversity of health condition amongst participants was deliberately sought by targeting online support groups with on social media platforms, including those for pain problems, mental health conditions, and individuals experiencing chronic fatigue, as well as including healthy people in the general population. Therefore, the derived factor structures and associated loading estimates should be robust for a wide variety of conditions (see comparison of the full dataset with those with non-zero pain below). Obviously, future studies in groups with specific health conditions are now needed.

1. The UDI items were derived from the Oswestry Disability Index and Neck Disability Index which were validated to measure disability due to pain. The item for pain was not included into the UDI. Was there a reason for this?

The item for pain was not included amongst the UDI items because of the desire to create a generic, condition-generic disability questionnaire based on ADLs.

2. There are other established generic disability tools such as Modified Barthel Index , or Katz Index of Independence in Activities of Daily Living. These are used in various types of conditions as well. How does the UDI compare with these tools? Thus far, the authors have discussed the comparison of UDI with GARS and very few other tools.

The motivation for creating the UDI was that there are surprisingly few generic self-reported disability questionnaires available. As mentioned in the existing introduction (pages 5-6):

“Surprisingly, relatively few generic self-reported disability questionnaires are currently available. Of those that are, there is either a focus on a specific aspect of disability, such as task independence [4], they incorporate complex ‘instrumental’ activities that may not be universally relevant [4-6], or they are very lengthy [5].”

Another reason for not adding more generic questionnaires to the survey (for example, the SF-36v2 and EQ-5D-5L were considered) was to not burden participants with too many questions and too lengthy a survey. Hence, the generic GARS and the pain-specific GCPS were deemed to be appropriate comparators in this initial evaluation.

To address the specific disability measures suggested here by Reviewer 2, the original Barthel Index [7] assesses ten common ADLs, including basic mobility (original version items: feeding, bathing, grooming, dressing, bowel, bladder, toilet use, transfers from bed-to-chair-and-back, mobility on level surfaces, stair negotiation). Neither original [7] nor modified versions [8] of the Barthel Index are self-reported questionnaires and are instead scored through observation by another person who is familiar with the ADLs being assessed. Similarly, the Katz Index of Independence in Activities of Daily Living is another tool that is scored by a third-party observer (items: bathing, dressing, toileting, transferring, continence, and feeding). This makes these in-person observation instruments impossible to test in a self-reported online survey, such as that used in the current study, but they are potential candidates for future validation studies that utilise a different methodology (i.e. not just self-reported questionnaires).

In addition, the ADLs incorporated in the Barthel Index and those of the Katz Index are typically only affected at the more severe end of the disability spectrum (e.g. following a stroke or other neurological disorder). Whilst some of these ADLs could feasibly be affected by painful conditions and health issues that give rise to fatigue, they are less likely to be markedly affected by mental health issues such as depression or anxiety (unless at the most extreme levels). By contrast, the UDI deliberately includes broader categories, such as ‘work and daily routine’, ‘social and recreational activities’ and ‘sleeping’, which are known to be affected by such conditions (even at mild or moderate severities) yet still has the capability to measure more severe disability.

3. The UDI was compared against GARS and GCPS. Comparison with GARS is suitable. However, I would like to know why the authors chose a pain-related scale for comparison. While pain is prevalent, not all disabilities are associated with pain. Patients with neurological disorders with loss of sensation may not experience pain but they still have disability. 

To a large extent, the range and choice of comparator disability scales was limited by their availability. By targeting independence of ADL performance, the GARS is condition-generic, even though a measure of independence is likely to be most relevant to the more severe end of the disability spectrum. By contrast, pain is very common in the general population [9], so it was absolutely essential to test the ability of the UDI to measure pain0related disability. Hence, the UDI was compared to a well-established pain-specific measure of disability. Given the decision to recruit a sample from the general population, there were very few pain-related disability questionnaires to choose from that were not anatomically restricted (e.g. not restricted to the lower back, neck, or shoulder, for example) and the Graded Chronic Pain Scale possessed several other virtues (it was brief, has good psychometric properties, and is the most widely used of those that met the other criteria). In future studies, the UDI will need to be compared to various condition/disease-specific instruments in relevant populations to further test the breadth/limits of its validity.

Therefore, did the authors select out participants who had pain for the analysis which compared the UDI and GCPS? Otherwise, those without pain might have discrepantly lower scores for GCPS. This could have affected the correlation between UDI and GCPS pain intensity. 

Many thanks for this suggestion. Pearson correlations were recalculated for the subset of participants with a non-zero current pain intensity score that had completed at least one item of the UDI (n=276). In this sensitivity analysis, correlations between disability measures were very similar to those of the full dataset (see tables below) and the correlation matrix for this subset (S6 Table) has been added to the Supporting Information.

Table 5. Correlation matrix of disability measures

 UDI 8 total UDI 7 total UDI 6 total GARS total GCPS disability

UDI 7 total 0.993 ***

(0.991, 0.994) 

UDI 6 total 0.987 ***

(0.985, 0.990) 0.995 ***

(0.994, 0.996) 

GARS total 0.827 ***

(0.791, 0.859) 0.840 ***

(0.807, 0.869) 0.834 ***

(0.800, 0.864) 

GCPS disability 0.735 ***

(0.673, 0.791) 0.722 ***

(0.655, 0.782) 0.719 ***

(0.651, 0.779) 0.631 ***

(0.556, 0.702) 

GCPS pain intensity 0.611 ***

(0.543, 0.677) 0.593 ***

(0.519, 0.659) 0.586 ***

(0.510, 0.652) 0.500 ***

(0.419, 0.580) 0.844 ***

(0.808, 0.876)

* p < 0.05

** p < 0.01

*** p < 0.001

S6 Table. Correlation matrix of disability measures for subset of participants with non-zero current pain intensity

 UDI 8 total UDI 7 total UDI 6 total GARS total GCPS disability

UDI 7 total 0.992 ***

(0.990, 0.994) 

UDI 6 total 0.986***

(0.982, 0.989) 0.994 *** 

(0.993, 0.995) 

GARS total 0.812 ***

(0.767, 0.850) 0.828 *** 

(0.788, 0.863) 0.820 *** 

(0.779, 0.857) 

GCPS disability 0.713 *** 

(0.640, 0.778) 0.706 *** 

(0.631, 0.773) 0.705 *** 

(0.628, 0.773) 0.614 ***

(0.528, 0.693) 

GCPS pain intensity 0.546 *** 

(0.458, 0.628) 0.536 *** 

(0.443, 0.619) 0.537 *** 

(0.448, 0.622) 0.460 *** 

(0.361, 0.559) 0.823 *** 

(0.781, 0.860)

* p < 0.05

** p < 0.01

*** p < 0.001

A line of text was added to the methods of the main text to describe this sensitivity analysis (page 16). A line of text was added to the methods of the main text to describe the results of this sensitivity analysis (page 16):

“In addition, as a sensitivity analysis, correlations were also assessed in the subset of participants with a non-zero ‘current’ pain intensity score (one of the GCPS items).”

In the results (page 25):

“In the subset of participants with non-zero current pain and at least one completed UDI item (n=276), point estimates (means) of these correlations were very similar to those of the full dataset (full results: the UDI8 was strongly correlated with both the GARS (r = 0.812 [0.767, 0.850]) and the GCPS disability (r = 0.713 [0.640, 0.778]). Full details of this sensitivity analysis are provided in supporting information.”

And in the discussion (page 30):

“The similarity in magnitudes of correlation coefficients for the full dataset with the subset of participants with non-zero current pain supports the validity of the UDI in capturing important aspects of disability, irrespective of the potential origins of this disability. It also indicates that the statistical models from which estimates were gained in this study are likely to be stable.”

Besides presenting the mean scores for their pain intensity in Table 1, do present the categorical results as well. e.g. How many n(%) had no pain, mild pain, moderate pain etc.

There is no agreed categorisation for descriptors of mild, moderate, severe pain intensity, beyond the preference of individual authors. This is partly because divisions in numerical pain rating scales are difficult to interpret [10]. In addition, the ‘characteristic pain intensity’ score of the GCPS (‘GCPS pain intensity), utilised here, is calculated as the mean of three different 0-10 numerical pain rating scales (current, worst, and perceived average), which makes it difficult to categorise meaningfully. This is why the standard deviation was used as an appropriate measure of pain intensity dispersion in Table 1. To offer categories with some validity, ‘Currently pain-free’ and ‘

---

## [Decision Letter · Decision Letter 1]

7 Dec 2023

PONE-D-23-31316R1Development and initial testing of a brief, generic self-reported disability questionnaire: the Universal Disability IndexPLOS ONE

Dear Dr. Evans,

Thank you for submitting your manuscript to PLOS ONE. After careful consideration, we feel that it has merit but does not fully meet PLOS ONE’s publication criteria as it currently stands. Therefore, we invite you to submit a revised version of the manuscript that addresses the points raised during the review process.

We look forward to receiving your revised manuscript.

Kind regards,

Mohammad Asghari Jafarabadi

Academic editor

Reviewers' comments:

Reviewer's Responses to Questions

**Comments to the Author**

1. If the authors have adequately addressed your comments raised in a previous round of review and you feel that this manuscript is now acceptable for publication, you may indicate that here to bypass the “Comments to the Author” section, enter your conflict of interest statement in the “Confidential to Editor” section, and submit your "Accept" recommendation.

Reviewer #2: (No Response)

Reviewer #3: (No Response)

2. Is the manuscript technically sound, and do the data support the conclusions?

Reviewer #2: Yes

Reviewer #3: Partly

3. Has the statistical analysis been performed appropriately and rigorously? 

Reviewer #2: Yes

Reviewer #3: No

4. Have the authors made all data underlying the findings in their manuscript fully available?

Reviewer #2: Yes

Reviewer #3: Yes

5. Is the manuscript presented in an intelligible fashion and written in standard English?

Reviewer #2: Yes

Reviewer #3: Yes

6. Review Comments to the Author

Reviewer #2: Thank you for addressing my comments in your response letter. The sensitivity analysis for pain has been incorporated into the manuscript. However, some of the other comments were answered in the response to reviewer but not incorporated into the manuscript. I apologise if this was not made clear in the previous review.

I accept the well-written rebuttals. I would suggest that some of the explanations should be included into the manuscript itself.

1. Include : Although pain was an item in the original Oswestry Disability Index and Neck Disability Index, the item for pain was not included in the UDI to ensure that the UDI was applicable to generic conditions.

2. Include a brief explanation that the UDI was aimed for patient self-report and hence different from measures such as the Barthel Index or Katz Index of Independence which were assessed by a third-party observer. It would also mean that the UDI can only be suitable for respondents who have adequate cognitive capacity to respond, as MBI and Katz Index can be used for patients with cognitive impairment.

3. Can the image quality for Figure 1 be improved?

Thank you.

Reviewer #3: ID: PONE-D-23-31316R1

Title: Development and initial testing of a brief, generic self-reported disability questionnaire: the Universal Disability Index

Thank you for providing a chance to review this manuscript.

Keywords: Keywords are generally 4-6, the current summary is not appropriate, please reconsider.

I am sorry to say that I have to reject the author's manuscript. On the one hand, it is puzzling that I did not receive any review comments from the authors in response, as well as no corrections in the manuscript for the problems I presented. On the other hand, RMSEA < 0.08 represents a fair fit. But in your manuscript, RMSEA > 0.1 indicates that there are problems with the model structure and that the model does not explain the observed data well.

Thank you and my best,

Your reviewer

7. PLOS authors have the option to publish the peer review history of their article (what does this mean?). If published, this will include your full peer review and any attached files.

Reviewer #2: **Yes: **Chai-Eng Tan

Reviewer #3: No

---

## [Author Response · Author response to Decision Letter 1]

16 Feb 2024

Reviewer #1:

No comments provided. Thanks you for your original review.

Reviewer #2:

Thank you for addressing my comments in your response letter. The sensitivity analysis for pain has been incorporated into the manuscript. However, some of the other comments were answered in the response to reviewer but not incorporated into the manuscript. I apologise if this was not made clear in the previous review.

I accept the well-written rebuttals. I would suggest that some of the explanations should be included into the manuscript itself.

1. Include : Although pain was an item in the original Oswestry Disability Index and Neck Disability Index, the item for pain was not included in the UDI to ensure that the UDI was applicable to generic conditions.

Thank you for clarifying. The following text has been added to the manuscript (p10) to explain why items were selected and others not:

Although pain is an item in both the ODI and NDI, a ‘pain’ item was not incorporated into the UDI because of the desire to create a condition-generic questionnaire based on ADLs. Instead, items related to everyday activities (e.g., walking, standing, lifting) or life situations (e.g., social, recreational, or work activities) were chosen so that the UDI aligned with the WHO’s ICF framework [5]. Similarly, …

2. Include a brief explanation that the UDI was aimed for patient self-report and hence different from measures such as the Barthel Index or Katz Index of Independence which were assessed by a third-party observer. It would also mean that the UDI can only be suitable for respondents who have adequate cognitive capacity to respond, as MBI and Katz Index can be used for patients with cognitive impairment.

Thank you for highlighting this important point. The introduction did state:

If the assessee has very severe levels of disability, in-person assessment may be the only option. However, people with all but the most severe levels of disability are often able to self-report their capabilities and restrictions through questionnaires.

Nevertheless, the following text has been added to the manuscript to make this important point more explicit (p33):

As with any self-reported questionnaire, the UDI requires that the respondent possess sufficient mental capacity to understand and respond to the questions. If this criterion is unmet, disability can only be assessed using third party observation, with instruments such as the Katz Index of Independence in Activities of Daily Living [6] or the Barthel Index [7]. On the other hand, third-party observations of ADLs are difficult to perform remotely, and currently require an observer who is trained to be familiar with the ADLs being assessed.

3. Can the image quality for Figure 1 be improved?

The original resolution of Figure 1 should have been adequate, even as a raster image. The Editorial Manager website PDF rendering tends to reduce this in the review documents. Please be reassured that the resolution of submitted images will be adequate for publication.

Thank you.

Thank you for taking time to review the manuscript again.

Reviewer #3: 

Thank you for providing a chance to review this manuscript.

Keywords: Keywords are generally 4-6, the current summary is not appropriate, please reconsider.

The keywords have been reduced to the following:

Disability; Questionnaire; Patient-Reported Outcome Measures (PROMS); Activities of Daily Living (ADLs); Factor analysis

I am sorry to say that I have to reject the author's manuscript. On the one hand, it is puzzling that I did not receive any review comments from the authors in response, as well as no corrections in the manuscript for the problems I presented. 

I can only apologise for this. However, your comments from the first review were not included in the information provided to me by the journal. Upon becoming aware (only in this second review) that a third reviewer had provided feedback, I immediately asked the journal office (by email) to send these comments to me (which were not evident within the EM website) and to let you know what had occurred. Unfortunately, despite waiting more than a month for a reply, I have not received any email response so must assume that this request has not been carried out by the journal. Obviously, I am happy to attend to these comments if you are happy to provide these again.

On the other hand, RMSEA < 0.08 represents a fair fit. But in your manuscript, RMSEA > 0.1 indicates that there are problems with the model structure and that the model does not explain the observed data well.

Thank you for this comment. As you will know, it is important to consider the collective evidence from multiple fit indices when performing CFA. The RMSEA values, while important to note (as we have in the results, limitations and the abstract), must be interpreted within the context of the entire model fit assessment and the theoretical framework guiding our study [1-3]. While mean RMSEA values from the three bootstrapped CFA models (0.12, 0.14, and 0.12) exceeded the conventional 0.08 threshold, all other fit indices – Chi square, CFI, TLI, SRMR, AIC, and BIC – all met their respective thresholds and thereby consistently indicated a good model fit. This broad agreement suggests robust model validity. Additionally, the lower bounds of our RMSEA confidence intervals approached and even crossed the 0.08 threshold (0.12 [0.08, 0.17], 0.14 [0.09, 0.19], 0.12 [0.06, 0.19]), suggesting an acceptable fit within statistical variability. RMSEA is known to be sensitive to various factors, including model specification and sample characteristics. Unfortunately, the nature of this variabliity can be complex and is not necessarily linear. Our study, with a moderate sample size of 334, could have resulted in variable RMSEA values. In addition, the deliberate sampling of heterogenous levels of disability might also have influenced RMSEA values. As mentioned in the ‘limitations’ section of the manuscript, testing of the UDI in more homogenous clinical samples is now necessary. Importantly, given the solid theoretical grounding of the UDI, based as it is on theoretically sound, well-tested and well-established existing disability questionnaires (the ODI and NDI), publication of appropriately investigated results that might be regarded as ‘negative’ by some is crucial for science [4, 5]. Hence, I have to strongly oppose the argument that the value of this single metric is a grounds for not publishing this study; a position that PLOS explicitly supports: https://everyone.plos.org/2020/04/06/filling-in-the-scientific-record-the-importance-of-negative-and-null-results/.

Thank you and my best,

Your reviewer

Thank you for taking time to review the manuscript.

References

1. Lai K, Green SB. The Problem with Having Two Watches: Assessment of Fit When RMSEA and CFI Disagree. Multivariate Behavioral Research. 2016;51(2-3):220-39. doi: 10.1080/00273171.2015.1134306.

2. Rigdon EE. CFI versus RMSEA: A comparison of two fit indexes for structural equation modeling. Structural Equation Modeling: A Multidisciplinary Journal. 1996;3(4):369-79. doi: 10.1080/10705519609540052.

3. Graham JM, Guthrie AC, Thompson B. Consequences of Not Interpreting Structure Coefficients in Published CFA Research: A Reminder. Structural Equation Modeling: A Multidisciplinary Journal. 2003;10(1):142-53. doi: 10.1207/S15328007SEM1001_7.

4. Ioannidis JP. Why most published research findings are false. PLoS Med. 2005;2(8):e124. Epub 20050830. doi: 10.1371/journal.pmed.0020124. PubMed PMID: 16060722; PubMed Central PMCID: PMCPMC1182327.

5. Young NS, Ioannidis JP, Al-Ubaydli O. Why current publication practices may distort science. PLoS Med. 2008;5(10):e201. doi: 10.1371/journal.pmed.0050201. PubMed PMID: 18844432; PubMed Central PMCID: PMCPMC2561077.

---

## [Decision Letter · Decision Letter 2]

26 Feb 2024

PONE-D-23-31316R2Development and initial testing of a brief, generic self-reported disability questionnaire: the Universal Disability IndexPLOS ONE

Dear Dr. Evans,

Thank you for submitting your manuscript to PLOS ONE. After careful consideration, we feel that it has merit but does not fully meet PLOS ONE’s publication criteria as it currently stands. Therefore, we invite you to submit a revised version of the manuscript that addresses the points raised during the review process.

We look forward to receiving your revised manuscript.

Kind regards,

Mohammad Asghari Jafarabadi

Academic Editor

PLOS ONE

Journal Requirements:

Reviewers' comments:

Reviewer's Responses to Questions

**Comments to the Author**

1. If the authors have adequately addressed your comments raised in a previous round of review and you feel that this manuscript is now acceptable for publication, you may indicate that here to bypass the “Comments to the Author” section, enter your conflict of interest statement in the “Confidential to Editor” section, and submit your "Accept" recommendation.

Reviewer #2: All comments have been addressed

Reviewer #3: (No Response)

2. Is the manuscript technically sound, and do the data support the conclusions?

Reviewer #2: Yes

Reviewer #3: Partly

3. Has the statistical analysis been performed appropriately and rigorously? 

Reviewer #2: Yes

Reviewer #3: Yes

4. Have the authors made all data underlying the findings in their manuscript fully available?

Reviewer #2: Yes

Reviewer #3: Yes

5. Is the manuscript presented in an intelligible fashion and written in standard English?

Reviewer #2: Yes

Reviewer #3: Yes

6. Review Comments to the Author

Reviewer #2: Thank you for addressing all the points in the review. Congratulations. I look forward to see the published version soon.

Reviewer #3: ID: PONE-D-23-31316R2

Title: Development and initial testing of a brief, generic self-reported disability questionnaire: the Universal Disability Index

Thank you for providing a chance to review this manuscript.

Detailed information:

Abstract

Background: Compared with the previous universal disability scale, what are the innovations of the scale in this paper?

Conclusions: The conclusion is not rigorous.

Introduction

1) Line 69, page 4: Explain the meaning and role of activities of daily living (ADLs).

2) Line 108-112, page 6: Please describe the innovation of this study.

Results

Table 1: The gender ratio was 2:1 so the sample was unevenly distributed.

Discussion

Overall:

The discussion section should be more specific about future research directions, including possible research questions, methods and areas of application.

7. PLOS authors have the option to publish the peer review history of their article (what does this mean?). If published, this will include your full peer review and any attached files.

Reviewer #2: **Yes: **Chai-Eng Tan

Reviewer #3: No

---

## [Author Response · Author response to Decision Letter 2]

29 Feb 2024

Reviewer comments

Reviewer #2:

Thank you for addressing all the points in the review. Congratulations. I look forward to see the published version soon.

Many thanks for your time and effort in providing constructive comments and suggestions.

Reviewer #3: 

ID: PONE-D-23-31316R2

Title: Development and initial testing of a brief, generic self-reported disability questionnaire: the Universal Disability Index

Thank you for providing a chance to review this manuscript.

Detailed information:

Abstract

Background: Compared with the previous universal disability scale, what are the innovations of the scale in this paper?

I view the UDI as novel (and needed) because there are very few, if any, existing generic disability questionnaires. The primary comparator questionnaire used in this study was the Groningen Activity Restriction Scale (GARS), but this measures independence during ADLs, which is similar to – but not the same as – self-reported ability in ADL performance. Many, if not most, disease-specific disability questionnaires use self-reported ability in ADL performance. Hence, the UDI is an attempt to fill this gap and was created using the wording of well-established disease-specific questionnaires that use concrete examples of ‘normal’ ADL performance (i.e. the Oswestry Disability Index [ODI] and the Neck Disability Index [NDI]).

The above was stated in the introduction as:

“Surprisingly, relatively few generic self-reported disability questionnaires are currently available. Of those that are, there is either a focus on a specific aspect of disability, such as task independence [19], they incorporate complex ‘instrumental’ activities that may not be universally relevant or might be outdated [11, 19, 34], or they are very lengthy [11].”

In the abstract, this was briefly summarised as: “Few generic self-reported disability questionnaires exist. Yet, these promise a broader and more comparable measure of disability than disease-specific instruments.”

Following the suggestion of Reviewer #3, I have attempted to increase the precision of this abstract wording to “A brief, generic self-reported disability questionnaire that promises a broader and more comparable measure of disability than disease-specific instruments does not currently exist.”

I have also added the following sentence (lines 226-228) to provide additional justification for using concrete examples within the UDI response options (as per the ODI and NDI):

“Concrete examples of ADL ability were chosen to ensure consistency between levels of reported (dis)ability, rather than asking respondents to gauge their recent ability compared to their historical levels, which would inevitably vary between individuals.”

Conclusions: The conclusion is not rigorous.

The current conclusion in the abstract is as follows:

“A brief, generic self-reported disability questionnaire was found to be valid and to possess good psychometric properties. The UDI has a single factor structure and either a 6-item, 7-item or 8-item version can be used to measure disability. For brevity and parsimony, the 6-item UDI is recommended, but further testing of all versions is warranted.”

I have to disagree with the general claim that this conclusion is “not rigorous” given that every component of this conclusion was carefully drawn from the results that are described in detail in the manuscript. Indeed, an attempt was made to be balanced with this conclusion, stating that further work should be done to evaluate the UDI. If Reviewer #3 has a concern with a particular component of the conclusion wording as it currently stands, then please do let me know and I will try to address this particular concern.

Introduction

1) Line 69, page 4: Explain the meaning and role of activities of daily living (ADLs).

Thank you for this suggestion. The sentence being referred to from the introduction is:

“According to the ICF, disability manifests as activity limitations and participation restrictions, thereby placing activities of daily living (ADLs) at the centre of disability assessment.”

ADLs have been a well-established concept in the disability literature since the 1950s, and are commonly used as an indicator of a person’s functional status. I have added a new reference in the existing sentence below that summarises the concept and history of ADLs for the reader’s benefit (line 97): https://pubmed.ncbi.nlm.nih.gov/29261878/

“According to the ICF, disability manifests as activity limitations and participation restrictions, thereby placing activities of daily living (ADLs) [6] at the centre of disability assessment.”

2) Line 108-112, page 6: Please describe the innovation of this study.

The lines referred to here are:

“With these considerations in mind, the primary aim of this study was to develop and evaluate the performance of a brief, generic self-reported disability questionnaire. This instrument seeks to address an existing gap in the literature by providing a versatile, yet rigorous, tool for assessing ADL-related disability across various demographic settings and health conditions.”

The justification for this aim is provided in the preceding paragraphs. Namely that: 1) a generic disability measure has several advantages over disease-specific measures; 2) there are few, if any, brief generic self-reported disability questionnaires in existence (instead, most existing generic disability measures rely on third-party in-person observation); 3) those few self-reported measures that do exist have issues such as (lines 105-108) “a focus on a specific aspect of disability, such as task independence [19], they incorporate complex ‘instrumental’ activities that may not be universally relevant or may be outdated [11, 19, 34], or they are very lengthy [11]”.

I have added the following sentence (lines 84-85) to clarify this further for readers:

“Despite this, most generic measures of disability are not self-reported and instead rely on in-person observation.”

Results

Table 1: The gender ratio was 2:1 so the sample was unevenly distributed.

There is little that one can (or should) do regarding the distribution of demographic features (age, sex, etc.) in a general population survey such as the one reported here. Since the analysis was largely descriptive, with no intervention (i.e. a clinical trial), such an ‘imbalance’ does not present a problem. Indeed, this female:male 2:1 ratio is typical of surveys conducted in the United Kingdom (UK), where the research was conducted, and examples of other large UK surveys with similar demographics were cited in the discussion to demonstrate this point. Similar ratios are also seen outside of the UK. The overall sample size (n=402) was sufficient for the purposes of the study. However, as pointed out in the discussion (lines 715-718), “future work should … facilitate the recruitment of under-represented groups to ensure the UDI items are appropriate and meaningful in these groups.”

Discussion

Overall:

The discussion section should be more specific about future research directions, including possible research questions, methods and areas of application.

A comprehensive description of what can be done with a generic disability questionnaire (which is a very broad topic) is beyond the scope of the paper, which is focused on describing the development of the questionnaire itself. In the short-term, the reliability and validity of the questionnaire needs further evaluation before it can be used in epidemiological studies and clinical trials, which is the long-term aim. Hence, several suggestions for future evaluation work are already provided in the discussion section, which should be sufficient:

713-717: “Hence, further studies will be required to create and test any translations of the UDI. Likewise, those with visual impairments would need the questions to be presented via a different medium (e.g., audio or braille) to provide a response. Hence, future work should look at different formats and facilitate the recruitment of under-represented groups to ensure the UDI items are appropriate and meaningful in these groups.”

733-734: “In future studies, this will need to be addressed by testing the UDI with individuals with a higher prevalence of more severe disability.”

Lines 738-740: “In future studies, this bias can be removed by the use of random item ordering, which is relatively straightforward to implement in online surveys.”

Lines 745-746: “Reliability studies should be a priority for future work.”

Many thanks to Reviewer #3 for their time and effort.

---

## [Decision Letter · Decision Letter 3]

19 Apr 2024

Development and initial testing of a brief, generic self-reported disability questionnaire: the Universal Disability Index

PONE-D-23-31316R3

Dear Dr. Evans,

We’re pleased to inform you that your manuscript has been judged scientifically suitable for publication and will be formally accepted for publication once it meets all outstanding technical requirements.

Kind regards,

Mohammad Asghari Jafarabadi

Academic Editor

PLOS ONE

Additional Editor Comments:

One or more of the reviewers has recommended that you cite specific previously published works. Members of the editorial team have determined that the works referenced are not directly related to the submitted manuscript. As such, please note that it is not necessary or expected to cite the works requested by the reviewer. Please note that we are accepting your manuscript at this time and do not require any further revisions as suggested by Reviewer 4 as these are not required to comply with our publication criteria.

Reviewers' comments:

Reviewer's Responses to Questions

**Comments to the Author**

1. If the authors have adequately addressed your comments raised in a previous round of review and you feel that this manuscript is now acceptable for publication, you may indicate that here to bypass the “Comments to the Author” section, enter your conflict of interest statement in the “Confidential to Editor” section, and submit your "Accept" recommendation.

Reviewer #4: All comments have been addressed

Reviewer #5: All comments have been addressed

2. Is the manuscript technically sound, and do the data support the conclusions?

Reviewer #4: Yes

Reviewer #5: Yes

3. Has the statistical analysis been performed appropriately and rigorously? 

Reviewer #4: Yes

Reviewer #5: Yes

4. Have the authors made all data underlying the findings in their manuscript fully available?

Reviewer #4: Yes

Reviewer #5: Yes

5. Is the manuscript presented in an intelligible fashion and written in standard English?

Reviewer #4: Yes

Reviewer #5: Yes

6. Review Comments to the Author

Reviewer #4: The article is of excellent quality. It is a very important topic for the scientific community. The small points I suggest improving are:

- Present an international overview of the theme in the introduction (what was done similarly in other countries, or even what was not done);

- Improve the practical implications of the study (How important is the study for society? How important is the study for professionals? How important is the study for the health system? Etc..

- I suggest reading the studies cited below, as I believe they can contribute to the introduction and/or discussion of the article and can be cited:

ANDRADE DO NASCIMENTO JÚNIOR, JOSÉ ROBERTO ; CAPRA DE OLIVEIRA, DAYANE ; LOPES PEREIRA MIRANDA DE ARAÚJO, VANESSA ; DIAS ANTUNES, MATEUS ; VICENTINI DE OLIVEIRA, DANIEL . Impacto da força muscular de membro inferior na capacidade funcional de idosas com osteoporose praticantes de hidroginástica. RBCEH. REVISTA BRASILEIRA DE CIÊNCIAS DO ENVELHECIMENTO HUMANO, v. 15, p. 33-45, 2018. Avaliable: https://seer.upf.br/index.php/rbceh/article/view/6422

Reviewer #5: The author took the comments received seriously and responded to them adequately. This way, accordinf to me, the paper is ready for publication.

7. PLOS authors have the option to publish the peer review history of their article (what does this mean?). If published, this will include your full peer review and any attached files.

Reviewer #4: **Yes: **Mateus Antunes

Reviewer #5: No
